🔓 | **Open Peer Review** | Bacteriology | Research Article

# Dairy manure, glyphosate, and antimicrobials (copper, streptomycin, and triazole) modulated the composition of antimicrobial resistance at the gene and microbial levels in a processing tomato field

Loic Deblais,[1] Gabrielle Derippe,[1] Madeline Horvat,[2] Sochina Ranjit,[1] Vincent Moulia,[1] Alejandra M. Jimenez Madrid,[2] Michael Kauffman,[1] Francesca Rotondo,[2] Melanie L. Lewis Ivey,[2] Sally A. Miller,[2] Gireesh Rajashekara[3]

**ABSTRACT** Intensive pesticide use drives antimicrobial resistance (AMR) in agriculture, yet the effects of specific practices remain poorly understood. This study evaluated the impact of dairy manure and agrochemicals (glyphosate, copper, streptomycin, and propiconazole) on the composition of culturable AMR bacteria (CARB), AMR genes (ARGs; $n = 87$), and the microbiome in a processing tomato field ($n = 64$ experimental plots). Glyphosate-treated plots harbored the lowest levels of CARB, but the highest prevalence of ARGs (especially *tetA*, *tetB*, OXA-50, and OXA-58) in the tomato leaves ($P < 0.05$). Manure-treated plots had the highest levels of CARB and ARGs in the soil and in tomato leaves (especially ACT-1, LAT, MIR, *aadA1,* and *aphA6*). The prevalence of multiple ARGs (IMP-12, ACT-1, DHA, MIR, MOX, OXA-58, OXA-60, *ermB*, *oprj,* and *oprm*) was lower in streptomycin- or propiconazole-treated plots compared to non-treated plots. Shifts in the soil and leaf microbiome correlated with changes in ARG composition, particularly aminoglycoside-, fluoroquinolone-, and beta-lactamase-associated genes. These findings show that dairy manure, glyphosate, and propiconazole significantly alter the tomato field microbiome and ARG landscape, indicating that fungicide and herbicide applications may contribute to AMR development and dissemination similar to conventional antibacterial agents in agricultural ecosystems.

**IMPORTANCE** Plant agricultural practices are commonly used by farmers to assure the yield and quality of crops; however, they are also associated with the emergence and dissemination of antimicrobial-resistant (AMR) pathogens. AMR is a critical concern in plant agriculture, as it can affect food safety, security, and sustainability. To combat this issue, it is critical to understand the impact of agricultural practices on AMR. Our study demonstrated that biological amendment (dairy manure) and pesticides (glyphosate, copper, streptomycin, and propiconazole) significantly exacerbated the AMR burden in the applied tomato field, which could increase the food safety risk of the fruit. Findings from this study will raise awareness among farmers, policymakers, and consumers, promote responsible and judicious use of antimicrobial agents in plant agriculture, and prioritize the development of sustainable practices to mitigate current and future AMR challenges.

**KEYWORDS** extended spectrum beta-lactamase, glyphosate, squeezed dairy manure, antimicrobial resistance, streptomycin, copper hydroxide, propiconazole, microbiome

**Peer Reviewer** Muhammad Afzal, South China Agricultural University College of Agriculture, Zhejiang, China

Address correspondence to Gireesh Rajashekara, gireesh2@illinois.edu.

The authors declare no conflict of interest.

See the funding table on p. 21.

The use of antimicrobial agents has significantly enhanced public health, food safety, and food security worldwide over the past several decades (1). However,

the widespread and often indiscriminate application of these agents has led to the emergence and dissemination of antimicrobial resistance (AMR) mechanisms in microorganisms. AMR now poses a critical threat to food systems, environmental health, and global public health (2, 3). Globally, AMR is estimated to affect over 10 million people annually and is associated with approximately 700,000 deaths (4, 5). In the USA alone, more than 2 million AMR-related infections are reported each year, resulting in at least 23,000 deaths (3). The economic burden is also substantial, with AMR-related infections costing the US government an estimated $2.2 billion in 2014—double the cost reported in 2002—and projections estimating a global economic loss of $100 trillion by 2050 if current trends continue (3). Therefore, there is a pressing need to identify AMR reservoirs and understand the factors leading to the development of AMR.

Agricultural practices, particularly the overuse and misuse of antimicrobials, are known contributors to the expanding AMR burden (1, 6). Notably, approximately 80% of antibiotics sold in the USA in 2011—equivalent to 13.6 tons—were used in livestock production (6). These practices have been shown to increase the abundance and diversity of AMR bacteria and resistance genes in animal manure, which is often applied to agricultural soils (1, 7–11). Although antibiotics such as streptomycin, oxytetracycline, oxolinic acid, and gentamicin are also used to manage bacterial diseases in crops (12–14), their application in plant agriculture remains limited, accounting for less than 1% of total antibiotic use (12, 13).

In contrast, pesticide use, particularly herbicides, has grown substantially. An estimated 2.7 million tons of pesticides were applied globally in 2012, including nearly 500,000 metric tons in the USA, with herbicides representing approximately 60% of all pesticides used in US plant agriculture (12, 13). Glyphosate, the most extensively used herbicide, was applied at a rate of 118,300 metric tons in the USA in 2017 and has documented antibacterial properties (13). Evidence suggests that the combination of glyphosate and antibiotics can accelerate the evolution of resistance in *Enterobacteriaceae* such as *Escherichia coli* and *Salmonella enterica* (13, 14). Furthermore, livestock manure—a reservoir of diverse and potentially pathogenic bacteria, including extended-spectrum beta-lactamase-producing Enterobacteriaceae—can amplify the risks when co-applied with such chemical inputs (15–17).

The persistence of antimicrobial residues in soil, crops, and water, often at sub-therapeutic levels, may further promote the development of resistance mechanisms (18, 19). Heavy metals such as copper hydroxide, which do not degrade in the environment like traditional antibiotics, are of particular concern. Copper-based antimicrobials are widely used across nearly all crop production systems—an estimated 4,600 tons of copper applied in the USA in 2015—and have been implicated in promoting resistance to clinically relevant antibiotics (20–23). Similar concerns exist with agricultural azoles like propiconazole, the third most commonly used triazole fungicide, which shares a mode of action with medical azoles and may drive cross-resistance in fungal pathogens (23, 24). Despite growing evidence on the individual impacts of antibiotics, herbicides, and heavy metals on AMR, an important research gap remains. Most studies investigate single inputs, whereas real-world farming systems involve multiple inputs applied simultaneously. The interplay of these factors is increasingly recognized as a driver of AMR proliferation in agricultural ecosystems, particularly in intensively managed crop systems such as tomato cultivation (25–29). Together, these findings underscore the urgent need to disentangle how the combined application of multiple agrochemical inputs acts to reshape soil microbiomes and enhance the risk of AMR transmission from farm environments to human and plant ecosystems.

Tomato farming provides a particularly significant model system for studying AMR dynamics. Tomatoes are one of the most widely consumed vegetables globally, often eaten raw, making them a direct exposure route for AMR bacteria and resistance genes from soil, irrigation water, or manure to humans. As a high-value crop frequently treated with herbicides, fungicides, and sometimes antibiotics, tomato fields represent a hotspot for interactions between diverse agricultural inputs and microbial communities.

The significance of studying AMR in tomato farming lies in its dual importance to safeguard food security and reduce risks to public health. Understanding how combined agricultural inputs shape soil and phyllosphere microbial communities, ARGs, and the potential transfer of resistance from the environment to the food chain is critical for designing sustainable farming practices and mitigating human exposure. In this study, we evaluated the combinatorial effects of squeezed dairy manure and commonly used agricultural inputs—glyphosate, streptomycin, copper hydroxide, and/or propiconazole—on the plant-associated microbial community in a processing tomato field. Specifically, we assessed the abundance of culturable bacteria resistant to copper, streptomycin, and beta-lactams (cefoxitin, cefepime, and meropenem), as well as the composition of ARGs. We further examined microbial community shifts in both soil and plant leaves (phyllosphere) to determine how these changes correlate with the enrichment and dissemination of ARGs. Environmental variables, including soil physicochemical properties, antimicrobial residues, and weather conditions, were also monitored to identify potential correlations with AMR emergence.

## MATERIALS AND METHODS

### Experiment design

An experimental field (33.5 × 94.4 m) located at The Ohio State University, CFAES-Wooster, in Wayne County, Ohio, USA (40°44'07.7"N 81°54'06.2"W) was used to determine the impact of agricultural practices on the culturable AMR bacteria (CARB), ARGs, and plant- and soil-associated microbiome of a processing tomato field. A total of 16 agricultural practice combinations were tested in this study. Each agricultural practice combination was randomized using a stratified strategy (four replicates per agricultural practice combination; total of 64 experimental plots; Fig. S1). Each experimental plot was composed of three rows of 20 plants each. Before planting, manure, glyphosate, or manure plus glyphosate were applied once to experimental plots (Table S1 and Fig. S2). After planting, experimental plots were treated on a weekly basis with streptomycin, copper, or propiconazole until harvest (Table S1 and Fig. S2). Non-treated experimental plots were used as a baseline to monitor the impact of the agricultural practices on CARB, ARGs, and the microbiome.

### Agricultural practices applied to the field

Squeezed dairy manure was manually applied at a rate of 11.2 t/ha (standard application in Northeast Ohio) on the frozen soil of selected experimental plots at time point 2 (TP2; Table S1 and Fig. S2). The manure was obtained from the CFAES-Wooster (previously the Ohio Agricultural Research and Development Center) dairy farm (The Ohio State University, Wooster, OH). Neither the cows nor their associated manure received any antibiotic treatments.

   A post-emergent application of the herbicide glyphosate (Roundup Weathermax Herbicide, Bayer Crop Science, St. Louis, MO; 48.8% potassium salt of glyphosate) was applied at a rate of 3.5 L/ha to the soil of selected experimental plots (1 day after TP3) using a Farmall 140 field sprayer (International Harvester, Lisle, IL, USA; Table S1 and Fig. S2). Antimicrobials (1.68 kg/ha of Kocide 2000 DuPont Crop [Houston, TX; 53.8% copper hydroxide]; 200 µg/L of Harbour [ADAMA; Raleigh, NC; 22.4% streptomycin sulfate]; 0.35 L/ha of Propimax EC [Indianapolis, IN; active ingredient: propiconazole]) were applied weekly during the growing season directly to the seedlings' foliage of the selected experimental plots using a handheld 11.3 L pressurized backpack sprayer with an 80° flat fan nozzle (276 kPa), as recommended by the manufacturers (Table S1). Weeds were removed manually from the experimental plots using appropriate personal protective equipment to avoid cross-contamination of the plots. Except for streptomycin, all agrochemicals were used according to the US Environmental Protection Agency registered labels. For streptomycin, off-label field applications were made for experimental purposes.

## Processing tomato production

Six-week-old "Peto 86" tomato seedlings (also known as "Hypeel 696" or "Perfect Peel"; Bayer, Germany) were transplanted into the field (density: one seedling every 0.3 meter) 92 days after manure application (TP6; Fig. S2) using a conventional Holland transplanter (Holland, MI). The field contained 3,840 tomato plants. Prior to transplanting into the field, seeds were hot water- and chlorine-treated to eradicate seedborne bacterial pathogens (30). The treated seeds were sown in 200-cell trays with Baccto professional grower's potting mix (Baccto, Houston, TX) and grown in a greenhouse for 5 weeks (min-max values: approximately 22°C–28°C and 20%–80% relative humidity; 12 h photoperiod). One week before transplanting, seedlings were grown outside the greenhouse for adaptation to environmental conditions. Tomato plants were overhead-irrigated in the greenhouse and drip-irrigated in the field using municipal water as needed.

## Soil and leaf sample collection and processing

A total of 3,840 samples (320 samples per time point per sample type) were collected throughout the nine time points (2,560 soil and 1,280 leaves; Fig. S2). Samples were collected 1 day before manure (TP1) and glyphosate (TP3) applications; 1 day before transplanting the seedlings into the field (TP4); and 1 day before the first, second, fifth, and sixth antimicrobial applications (TP6–TP9, respectively). Manure samples and seedlings (TP2 and TP5, respectively; $n = 8$ samples each) were also collected. A total of five soil and leaf samples were collected per experimental plot ($n = 320$ samples per time point). Samples were pooled together by sample type and by experimental plot ($n = 64$ pooled samples; four reps per agricultural practice combination tested) into a 600 mL Whirl-Pak bag (Nasco, Fort Atkinson, WI). Different locations and seedlings within the experimental plot were collected between time points. Soil and leaf samples were collected from the same vicinity to facilitate the comparison of the data between sample types. The soil was sampled by collecting the first five centimeters of soil within a radius of 30 cm around the selected seedling using 1.27 cm diameter PVC pipes (Sliver-Line Plastics, Asheville, NC) to collect the majority of the bacterial population of the soil (31). The oldest leaf (healthy and non-senescent) of the selected plants was collected into a 600 mL Whirl-Pak bag. All samples were stored in a Styrofoam box containing ice packs until processing in the lab on the same day.

The samples were weighed before adding 50 mL of peptone water (Sigma-Aldrich, St. Louis, MO) into each Whirl-Pak bag. Soil samples were manually homogenized for 1 min, and leaf tissues were macerated before aliquoting them for microbiology and molecular approaches. A second set of unprocessed soil samples was collected as described above to determine the chemical properties of the soil, as described below.

## Quantification of culturable bacteria resistant to copper, streptomycin, cefepime, cefoxitin, and meropenem in soil and leaves

Homogenized soil and leaf samples were 10-fold serially diluted and plated on MacConkey agar amended with a determined concentration of antimicrobials (200 µg/mL copper [Fisher Scientific, Hampton, NH]; 200 µg/mL streptomycin [Sigma-Aldrich, St. Louis, MO]; 4 µg/mL cefepime [Sigma-Aldrich]; 8 µg/mL cefoxitin [Sigma-Aldrich]; 1 µg/mL meropenem [Sigma-Aldrich]). Inoculated plates were incubated at room temperature for up to 5 days.

## DNA extraction from soil and leaf samples

Total genomic DNA from the soil and leaf samples was extracted using the PowerSoil DNA Isolation Kit (Qiagen, Germantown, MD) and the Wizard Genomic DNA Purification Kit (Promega, Fitchburg, WI) respectively, as indicated by the manufacturers. The DNA was then resuspended in 100 µL of RNase-free sterile water (Qiagen, Germantown, MD). Additional DNA-cleaning steps using Genomic DNA Clean & Concentrator Kit (Zymo

Research, Irvine, CA) were performed if the DNA quality was insufficient (260/280 ratio < 1.8) or due to the presence of insoluble black precipitate (high polyphenol and polysaccharide residues) (32). DNA extraction from the peptone water used during the processing of the field samples was used as background-noise control for the molecular and sequencing approaches. The quality and concentration of the extracted DNA were verified using NanoDrop 2000/2000c Spectrophotometer (ND-2000, Thermo Scientific, Waltham, MA). DNA samples were stored at −20°C until further analyses.

## Composition of antibiotic resistance genes in soil and leaf samples using real-time PCR

The Antibiotic Resistance Genes Microbial DNA qPCR Array (Qiagen, Germantown, MD) was used to determine the composition of antibiotic resistance genes in the soil and leaf samples. Details concerning the 16 antibiotic resistance groups studied and the sensitivity of the primer pairs ($n = 87$) used are described in Table S2. Approximately 10.5 ng of DNA was used per well, as recommended by the manufacturer. The following qPCR settings were used: denaturation at 95°C for 5 min, then 40 cycles with 95°C for 15 s, 62°C for 30 s, and 72°C for 15 s. One plate was run with sterile water to determine the fluorescence background noise measured for each primer to assure the quality of the hits (i.e., positive amplification of the designated gene).

## Microbiome composition in field samples using MiSeq sequencing and QIIME 2

A total of 50 ng DNA was used to generate amplicons of the 16S rRNA (V4-V5 region; primers 515F and 926R) (33) using Phusion High-Fidelity PCR Kit (New England Biolabs Inc., Ipswich, MA). PCR products were cleaned using AMPure XP PCR Kit (Beckman Coulter Inc., Beverly, MA), and sequenced using Illumina MiSeq 300-base paired-end kit, as described previously (34, 35). Quality control and processing of the raw reads were performed using FastQC (Babraham Bioinformatics, Cambridge, UK), and Trimmomatic. Raw, single-end sequence reads were processed using QIIME2 v.2020.11 (36). The DADA2 plugin was used to truncate reads at 280 bp and to denoise the single-end reads (37). Soil samples were filtered at a sequencing depth of 15,000 reads, while leaf samples were filtered at a depth of 950 reads due to the lower amount of reads in the samples. Taxonomic assignment was performed using QIIME2 and the latest SILVA reference database (version 138.1; 99% homology cut-off) (38). Reads that were identified as eukaryotic, mitochondrial, or chloroplast were filtered out.

## Estimation of the total bacterial population in soil using flow cytometry

The extraction of the bacterial cell from the soil particles was performed as previously described (39). Briefly, 5 g of soil was diluted in 45 mL of 0.85% NaCl, vortexed for 5 min at full speed, and centrifuged at 130 $g$ for 5 min. The supernatant was filtered with a BD Falcon 40 µm cell strainer (Falcon, Bedford, MA) and centrifuged at 4,700 $g$ for 12 min. The pellet was resuspended in 1 mL of 1× phosphate-buffered saline (PBS; pH 7.4). An aliquot (15 µL) of the resuspended pellet was transferred into a 1.5 mL tube containing 985 µL of 1× PBS and 15 µL of 16% formaldehyde (Sigma-Aldrich, St. Louis, MO), mixed three times by inversion, and incubated at room temperature for 15 min. The rest of the suspension was stored at −80°C. The fixed cells were stained with 5 µL SYBR Green I (200× in DMSO; Sigma-Aldrich, St. Louis, MO) and incubated in the dark at 4°C for 20 min. Flow cytometry analyses were performed using BD FACSAria IIU machine equipped with a 488 nm argon laser. Unstained samples and 0.85% NaCl buffer were used as controls. At least 2.5 million particles per sample were measured to determine the number of SYBR Green-stained organisms per gram of soil. The concentration of microorganisms (cell size <40 µm) in the soil was estimated by normalizing the number of SYBR Green-positive particles based on the volume input into the flow cytometry machine.

## Streptomycin residues quantification in soil using liquid chromatography

The streptomycin residue in the soil was determined using cryogenization (Labconco 4.5 Liter Freeze Dry System Machine; Labconco, Kansas City, MO), solid-phase extraction (SPE; Waters, Milford, MA), and liquid chromatography combined with mass spectrometry (LC-MS; Bruker 15T SolariXR instrument), as previously described (40–42). Briefly, soil samples were freeze-dried ($-47°C$, $1 \times 10^3$ mbar) overnight in the dark and then sieved (0.6 mm diameter). One gram of dried, sieved soil was transferred in a 50 mL tube containing 20 mL of McIlvaine buffer (pH 4) supplemented with 0.05% $Na_2EDTA$, homogenized at room temperature for 20 min at 200 rpm, and centrifuged for 15 min at 3,700 $g$. The pellet was resuspended with 20 mL of McIlvaine buffer (pH 4) supplemented with 0.05% $Na_2EDTA$, and the steps described above were repeated. The supernatant ($n$ = 20 mL per sample) was filtered (0.2 µm), then passed through a cartridge (6 mL/150 mg; OASIS WXC SPE, Waters Milfort, MA), and resuspended in 1.2 mL of 100% methanol. The product of the elution was dried for 3 h using SpeedVac and resuspended in 100 µL of 10% methanol supplemented with 1% formic acid and 100 ppm rapamycin (a sample containing only methanol, formic acid, and rapamycin was used as a blank control for the LC-MS/MS analysis). Five microliters of the final product was injected per run for LC-MS/MS analysis. Each sample was analyzed three times. A standard curve was created by amending non-treated soil samples collected from the same plots with a determined concentration of streptomycin sulfate (0.0, 0.1, 0.25, 0.5, 1.0, 10.0, or 100.0 ng of streptomycin per gram of soil) and processed as described above.

## Analyses of soil chemical properties

Fifty grams of soil sample were freeze-dried ($-47°C$, $1 \times 10^3$ mbar) overnight in the dark, sieved (0.6 mm diameter), and shipped on ice to Brookside Laboratories Inc. (New Bremen, OH) to analyze the soil chemical properties. A total of 25 parameters were analyzed (Ref: S001AN; https://www.blinc.com/agricultural-soils). Elements (P, Mn, Zn, B, Cu, Fe, Al, S, Ca, Mg, K, and Na) were extracted and measured using the Mehlich III method. The pH of the soil was determined using Shoemaker, McLean, and Pratt (SMP) and Sikora buffer methods.

## Statistical analyses

Bacterial counts obtained from the direct plating of the field samples were log-transformed (Log CFU per gram of tissue). The normality and the homogeneity of the data were confirmed using Shapiro and Bartlett tests, respectively. A one-way and two-way analysis of variance combined with a Tukey test was used to detect significant differences in CARB load between the agricultural practice combinations within and between time points, respectively. The bacterial counts were expressed as average plus intervals of confidence (IC95%). A row-wise method and Pearson Product-Moment Correlation analysis were performed to detect correlations between different sets of continuous variables (i.e., AMR bacterial counts, total bacterial counts, real-time PCR data, soil properties, weather conditions [from the Ohio Agricultural Research Development Center, Wooster weather station], total microorganism population in the soil, and antimicrobial residues in the soil). A cumulative sum of the AMR data (bacterial quantification and AMR gene prevalence) was used to assess the overall impact of agricultural practice combinations on the AMR burden. Two-way clustering, principal component, and discriminant analyses were performed to estimate the variability observed between the agricultural practice combinations for a specific set of data. The diversity and abundance of the AMR genes was analyzed using Venn diagrams (43). All the statistical analyses were performed using JMP Pro 16 (SAS, Cary, NC) and R Studio (Boston, MA). A pairwise Wilcoxon test was performed to identify differences in relative abundance in culturable AMR bacteria in the soil (percentage of culturable AMR bacteria compared to the total microbial population) between agricultural practice combinations for a given time point. Alpha diversity was analyzed using Shannon (richness) and Faith's PD (phylodiversity) indices. A Kruskal-Wallis' test was used to identify differences in

alpha diversity. A permutational multivariate analysis of variances was used to identify differences in beta diversity (unweighted and weighted UniFrac) based on Bray-Curtis distance matrices. An analysis of the composition of microbes was used to identify differences in relative abundance at the phylum and species levels (44). Furthermore, a Bootstrap Forest analysis (false discovery rate, equivalence test, and Huber M-estimation) was used to identify only OTUs large enough to be of pragmatic interest and reduce the impact of outliers on the statistical differences (JMP Pro 16; SAS, Cary, NC). The statistical analyses mentioned above focused on the differences observed between the sample types and agricultural practice combinations for a designated time point. $P$-values less than 0.01 were considered significant.

## RESULTS

### Impact of manure and glyphosate on the abundance of CARB in the soil and tomato leaves collected before the growing season (TP1–TP6)

#### Baseline CARB levels and manure impact

Initial soil analysis (TP1) showed high and spatially uniform populations of CARB, particularly those resistant to copper (4.64-log), cefoxitin (4.62-log), and cefepime (4.40-log), with lower populations resistant to meropenem (4.16-log) and streptomycin (2.37-log; $P < 0.01$; Fig. 1A). Manure collected before field application (TP2) exhibited significantly higher CARB levels, notably for copper (6.53-log) and streptomycin (5.40-log), than soil at TP1 ($P < 0.01$; Fig. 1B). However, by TP3 (86 days post-application), manure-treated soil (M group) showed no significant difference in CARB levels compared to untreated plots (C group; Fig. 2A).

#### Glyphosate effects on soil CARB

Glyphosate treatment, with or without manure (G and GM groups), significantly increased total CARB levels in soil by TP4 (3 days post-glyphosate; 89 days post-manure), reaching 6.59-log versus 6.36-log in untreated plots ($P < 0.05$; Fig. 2B). Increases were primarily driven by copper-resistant (6.24-log vs 5.91-log) and meropenem-resistant (5.92-log vs 5.60-log) bacteria ($P < 0.05$; Fig. S3A). Co-treatment (GM group) further elevated copper- (6.27-log) and cefepime-resistant (5.88-log) bacteria compared to manure alone (M group: 5.91-log and 5.48-log, respectively; $P < 0.05$; Fig. 2B and S3B). These glyphosate-associated increases were transient, as CARB levels across groups converged by TP6 (10 days post-glyphosate; 96 days post-manure). However, GM-treated plots still showed higher levels of cefepime- (5.58-log) and cefoxitin-resistant (5.89-log) bacteria than glyphosate-only plots (G group: 5.22-log and 5.56-log, respectively; $P < 0.05$; Fig. 2C and S3B).

#### CARB populations in tomato leaves

By TP6 (6 days post-transplantation), glyphosate-treated plots (G and GM) had elevated streptomycin-resistant bacteria in leaves (4.55-log) compared to untreated plots (C and M: 3.86-log; $P < 0.05$; Fig. S3A). Copper-resistant bacteria were also more abundant in leaves from glyphosate-only plots (5.97-log) than in controls (5.20-log; $P < 0.05$; Fig. 3A and S3B). Seedling analysis (Fig. S4) revealed significantly lower CARB levels than in field samples ($P < 0.05$), with most CARB (except streptomycin-resistant) concentrated in roots over foliage ($P < 0.05$).

### Impact of agricultural practices on the abundance of CARB present in the soil and tomato leaf tissues collected during the growing season (TP6 to TP9)

Analysis of samples collected between TP6 and TP9 revealed that the abundance of CARB in both soil and leaf tissues was primarily influenced by the application of dairy manure and glyphosate prior to the growing season. In contrast, foliar pesticide treatments (copper hydroxide, streptomycin, and propiconazole) applied during the growing

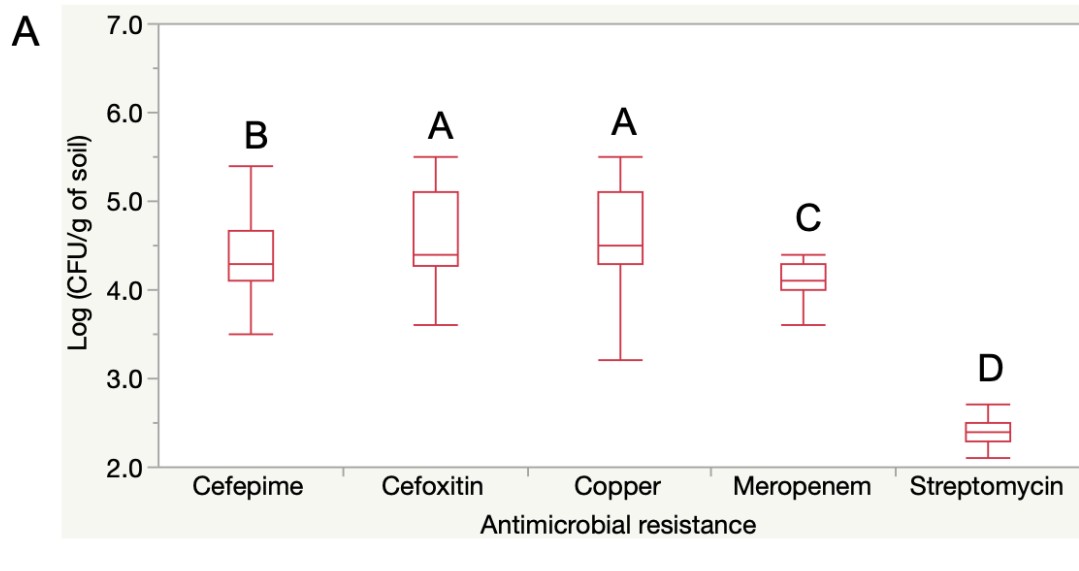

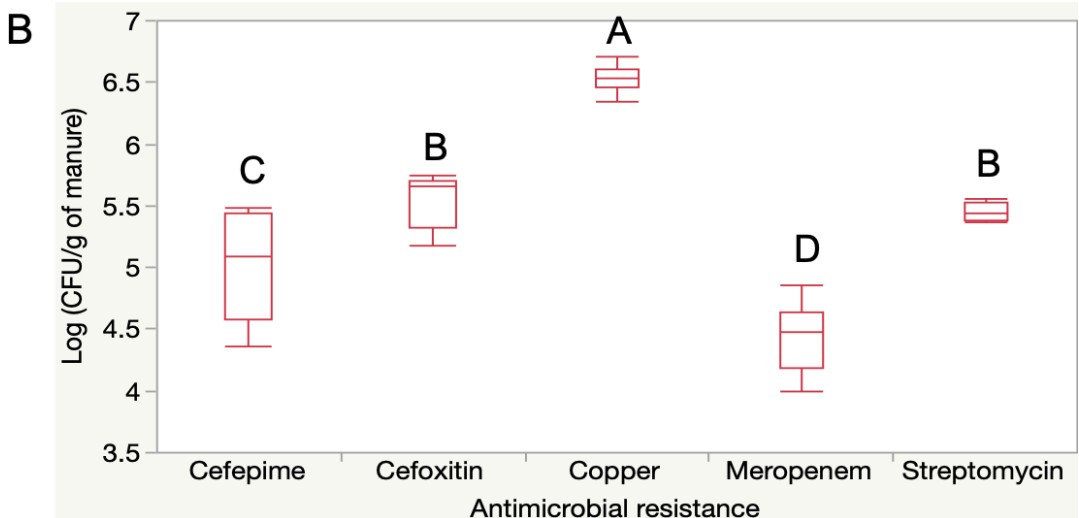

**FIG 1** Bacterial load of CARB in soil and dairy manure. (A) Antimicrobial-resistant bacterial profile from the soil collected on TP1 (1 day before manure application). (B) Antimicrobial-resistant bacterial profile of the manure applied in the field on TP2. Culturable bacteria resistant to 200 µg/mL copper, 200 µg/mL streptomycin, 4 µg/mL cefepime, 8 µg/mL cefoxitin, and 1 µg/mL of meropenem determined by direct plating on MacConkey agar. $N = 64$ pooled soil samples. Letters represent statistically significant categories ($P < 0.05$).

season had a secondary, modifying effect. These trends were more pronounced in leaf tissues (Fig. 3B through D; Fig. S3A through C) than in soils (Fig. 2D through F; Fig. S3A through C).

At TP7 (6 days after the first antimicrobial application), leaf tissues from plots treated with glyphosate (G and GM groups) harbored significantly higher levels of streptomycin-resistant bacteria (5.99-log [5.74–6.24] CFU/g) compared to untreated plots (C and M groups; 5.65-log [5.44–5.85] CFU/g; $P < 0.05$; Fig. S3A). Glyphosate-treated plots also clustered together (cluster α; Fig. 3B) and exhibited elevated levels of copper- and cefoxitin-resistant bacteria. This effect was significant in the G group alone (copper: 7.62-log [7.26–7.98]; cefoxitin: 7.45-log [7.00–7.91] CFU/g), compared to controls (6.77-log and 6.80-log, respectively; $P < 0.05$; Fig. S3B).

By TP8 (6 days after the fourth antimicrobial application), these trends intensified. All glyphosate-only plots (G group) clustered tightly (cluster β; Fig. 3C) and exhibited significantly lower CARB levels across all five tested antimicrobials compared to other treatment groups ($P < 0.05$). Conversely, most manure-treated plots (M and GM groups)

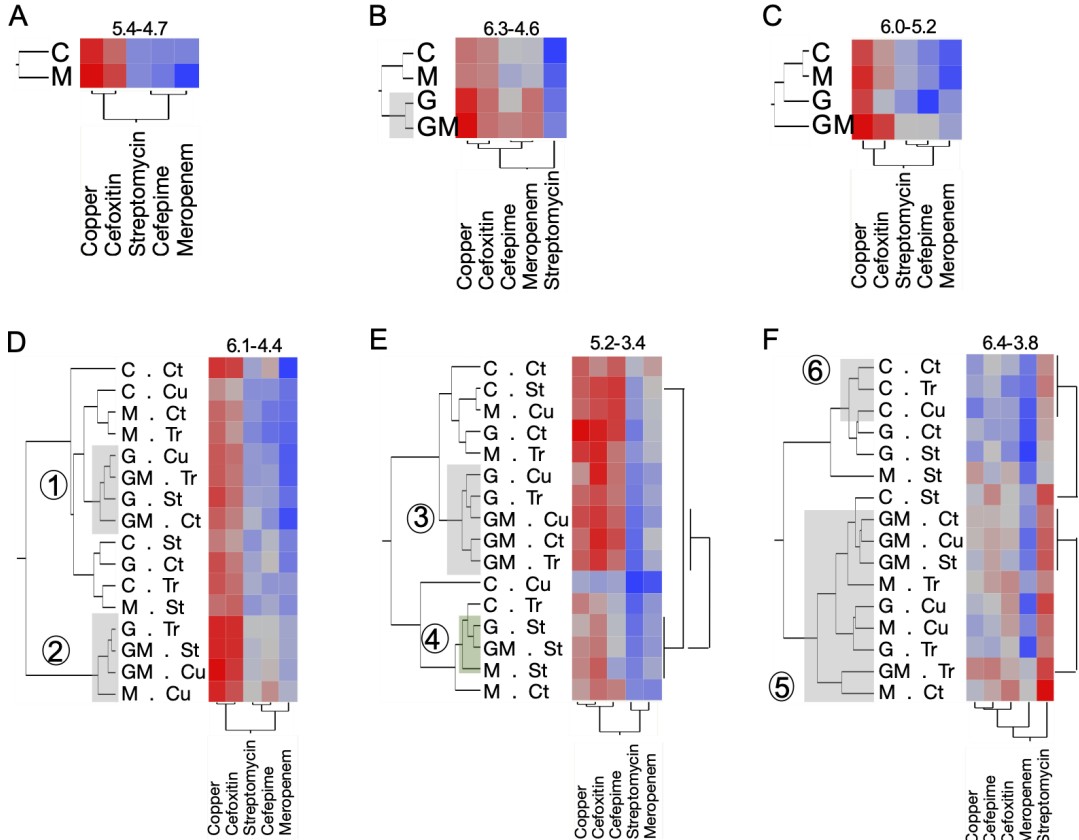

**FIG 2** Antimicrobial-resistant bacterial population differences in soil samples collected between TP3 and TP9. (A–C) Bacterial counts from the soil collected on TP3 (86 days post-manure application), TP4 (3 days post-glyphosate; 89 days post-manure application), and TP6 (10 days post-glyphosate; 96 days post-manure application). (D–F) Bacterial counts from the soil collected on TP7 (6 days after the first antimicrobial application), TP8 (6 days after the fourth antimicrobial application), and TP9 (6 days after the fifth antimicrobial application). Quantification of culturable antimicrobial-resistant bacteria was performed by direct plating on MacConkey agar amended with antibiotics (200 µg/mL copper, 200 µg/mL streptomycin, 4 µg/mL cefepime, 8 µg/mL cefoxitin, and 1 µg/mL of meropenem). $N = 64$ samples per time point. Heatmaps were created using the means of the biological replicates (32 reps at TP3; 16 reps at TP4 and 6; 4 reps at TP7-TP9). The color code was normalized across the five antibiotics for each time point. The increase in bacterial counts is proportional to the color code, from blue to red. Range (min-max log [CFU/g of leaf]) is indicated above each heatmap. Circled numbers represent clusters (plots with similar treatments) identified based on the hierarchical clustering analysis. G, GM, and M: plots to which glyphosate and/or manure were applied; C: non-treated plots. Cu, St, and Tr: plots to which copper hydroxide, streptomycin, or propiconazole were applied, respectively; C and Ct: non-treated control rows and plot, respectively.

formed a separate cluster (cluster Ω) characterized by significantly higher CARB levels, excluding streptomycin resistance (Fig. S3C). Similar patterns persisted at TP9 (Fig. 3D; Fig. S3B).

Parallel trends were observed in soil samples. At TP7, 87% of glyphosate-treated plots (G and GM groups) clustered into two groups (clusters 1 and 2; Fig. 2D), with cluster 2 representing the highest CARB levels. These clustering patterns weakened at TP8 (62.5% in cluster 3; Fig. 2E), but strengthened again at TP9, when most manure-treated plots (M and GM) formed cluster 5 and exhibited significantly higher levels of copper-, cefoxitin-, and meropenem-resistant bacteria compared to non-manure plots ($P < 0.05$; Fig. S3C). Notably, nearly all untreated control plots (C group), except those exposed to streptomycin (C.St group), clustered together at TP9 (cluster 6; Fig. 2F) and displayed the lowest CARB levels.

Copper applications also shaped CARB profiles. At TP7, most leaf samples from copper-treated plots (except those in the G.Cu group) clustered together (cluster λ; Fig. 3B). G.Cu plots exhibited significantly higher CARB levels compared to other copper-treated groups (Fig. 3B; Fig. S5A), but by TP9, the reverse was true—G.Cu leaves had significantly lower CARB levels (Fig. 3D; Fig. S5B). In soil, the C.Cu plots (copper-treated

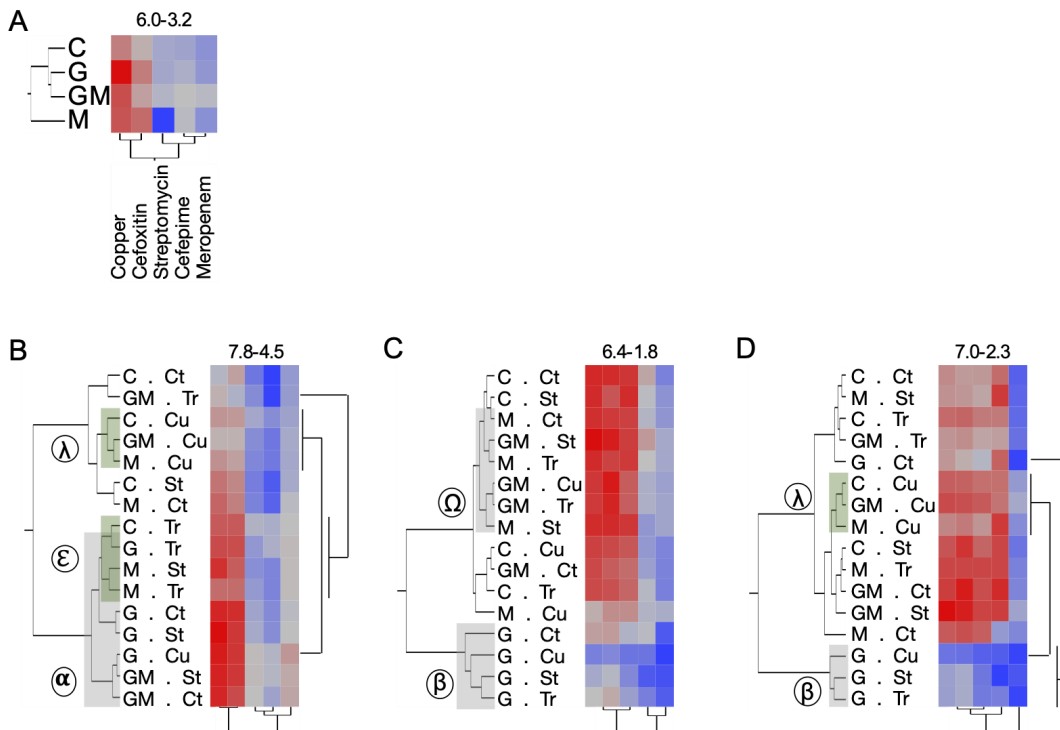

**FIG 3** Antimicrobial-resistant bacterial population differences in the leaf samples collected between TP6 and TP9. (A) Bacterial counts from leaves collected on TP6 (10 days post-glyphosate; 96 days post-manure application). (B–D) Bacterial counts from the leaf samples collected on TP7 (6 days after the first antimicrobial application), TP8 (6 days after the fourth antimicrobial application), and TP9 (6 days after the fifth antimicrobial application). Quantification of culturable antimicrobial-resistant bacteria was performed by direct plating on MacConkey agar amended with antibiotics (200 µg/mL copper, 200 µg/mL streptomycin, 4 µg/mL cefepime, 8 µg/mL cefoxitin, and 1 µg/mL of meropenem). N = 64 samples per time point. Heatmaps were created using the means of the biological replicates (16 reps at TP6; four reps at TP7–TP9). The color code was normalized across the five antibiotics for each time point. The increase in bacterial counts is proportional to the color code, from blue to red. Range (min-max log [CFU/g of leaf]) is indicated above each heatmap. Circled symbols represent clusters (plots with similar treatments) identified based on the hierarchical clustering analysis. G, GM, and M: plots to which glyphosate and/or manure were applied; C: non-treated plots. Cu, St, and Tr: plots to which copper hydroxide, streptomycin, or propiconazole were applied, respectively; C and Ct: non-treated control rows and plot, respectively.

but no manure or glyphosate) consistently showed the lowest CARB levels across all treatment combinations (Fig. 2D through F).

Propiconazole and streptomycin applications also influenced microbial community structure. At TP7, all leaf samples treated with propiconazole (except GM.Tr plots) clustered together (cluster ε; Fig. 3B). Similarly, soil samples from streptomycin-treated plots (except those lacking glyphosate and manure) clustered at TP8 (cluster 4; Fig. 2E), indicating that individual pesticide treatments modulated the effect of pre-season amendments.

## Effects of dairy manure and glyphosate on soil total microbial load

Between TP3 and TP9, total microbial populations in soil (cell size <40 µm) ranged from 9.10-log to 9.57-log SYBR Green-positive particles per gram (Fig. S6A). Significant differences were observed at TP6 and TP7 (Fig. S6B and C), where plots treated with both manure and glyphosate (GM group) had lower microbial loads than those treated with glyphosate (G group) or manure alone (M group; $P < 0.05$). Specifically, the GM group recorded 9.02-log (TP6) and 9.31-log (TP7) versus 9.43-log and 9.57-log (G group), and 9.42-log and 9.70-log (M group), respectively.

Despite reduced total microbial counts, the GM group showed significantly higher relative abundance of CARB at TP7—2.2-fold (copper), 2.6-fold (streptomycin), and 2.8-fold (cefoxitin) higher—compared to the other treatments ($P < 0.05$; Fig. S6D). Environmental factors also influenced CARB levels and patterns across soil and leaf samples from TP1 to TP9, independent of treatment type (Fig. S7). Additional details concerning correlation data between environmental conditions and CARB are shown in Fig. S7.

## Impact of agricultural practices on the ARG composition in soil and tomato leaves

A total of 48 of 87 ARGs from 13 antimicrobial classes were detected in 51 soil and 48 leaf samples across TP1–TP9 (Fig. S8). Most ARGs (77%) were shared between soil and leaf samples. Manure and/or glyphosate treatments (M, G, GM groups) significantly increased the prevalence of certain ARG classes in leaves ($P < 0.05$; Table 1). For instance, class C beta-lactamase genes were more prevalent in the GM and M groups (91%) vs the G group (45%), and tetracycline efflux pumps were more frequent in G (82%) and M (73%) vs C (25%) and GM (36%) groups.

Soils from M and GM plots showed a non-significant trend of higher ARG prevalence, particularly tetracycline and erythromycin resistance (Table 1). In contrast, leaves from AM-treated plots (Cu, St, Tr) generally showed a lower, though non-significant, ARG prevalence compared to untreated plots (Ct group), except for class A beta-lactamase genes, which were significantly more prevalent in Cu and St groups (>27%) vs Ct and Tr (0%; $P < 0.05$; Table 2).

During the pre-growing season (TP1–TP6), ARG diversity increased independently of manure or glyphosate application. Ten ARGs from five AMR classes were detected in initial soil samples (TP1; Table S3A), and similar genes were present in manure (TP2), often at higher levels (2.3-fold). However, manure-specific genes (e.g., ereB, VEB, OXA variants, tetB) were not detected in treated plots (TP3 or TP6). Leaf samples at TP6 had fewer ARGs ($n = 10$) compared to soil ($n = 18$; Fig. 4), with only five shared genes—these were abundant in leaves but scarce in soil (Fig. S8A).

**TABLE 1** Effect of dairy manure and glyphosate applications on antimicrobial-resistant gene prevalence in soil and tomato leaves between TP4 and TP9[a]

| AMR class | Leaf | | | | Soil | | | |
|---|---|---|---|---|---|---|---|---|
| | C | G | GM | M | C | G | GM | M |
| Aminoglycoside-resistance (5) | 50 | 45 | 73 | 55 | 100 | 100 | 100 | 100 |
| Beta-lactam resistance (1) | 13 | 27 | 0 | 0 | 0 | 9 | 0 | 9 |
| Class A beta-lactamase (22) | 13 | 36 | 9 | 9 | 9 | 9 | 18 | 18 |
| Class B beta-lactamase (9) | 0 | 18 | 18 | 18 | 100 | 91 | 100 | 100 |
| Class C beta-lactamase (11) | 63 | 45 | 91 | 91 | 91 | 100 | 100 | 100 |
| Class D beta-lactamase (13) | 38 | 64 | 36 | 36 | 100 | 100 | 100 | 100 |
| Erythromycin resistance (1) | 0 | 0 | 0 | 9 | 9 | 9 | 27 | 18 |
| Fluoroquinolone resistance (11) | 0 | 27 | 36 | 27 | 64 | 45 | 55 | 73 |
| IgG binding protein A precursor (1) | 0 | 0 | 0 | 0 | 0 | 0 | 0 | 0 |
| Macrolide lincosamide streptogramin_b (1) | 88 | 91 | 91 | 91 | 100 | 100 | 100 | 91 |
| Multidrug resistance efflux pump (2) | 13 | 9 | 0 | 0 | 9 | 0 | 0 | 0 |
| Panton-Valentine leukocidin chain F (1) | 0 | 0 | 0 | 0 | 0 | 0 | 0 | 0 |
| *Staphylococcus aureus* (1) | 38 | 55 | 64 | 36 | 9 | 9 | 0 | 18 |
| Tetracycline efflux pump (2) | 25 | 82 | 36 | 73 | 45 | 27 | 55 | 45 |
| Vancomycin resistance (2) | 0 | 0 | 0 | 9 | 0 | 0 | 0 | 0 |
| Cumulative score | 338 | 500 | 455 | 455 | 636 | 600 | 655 | 673 |

[a]Number in parentheses refers to the number of ARGs studied; ARG: antimicrobial resistant gene; G, GM, and M: plots to which glyphosate and/or manure were applied; C: non-treated plots. Cumulative score: sum of the prevalence for the given treatment group (max value = 1,500). $N = 13$ soil and 12 leaf samples per group. TP4: after glyphosate application; TP6: before the first application of the antimicrobials; TP7–9: after the application of antimicrobials on a weekly basis (additional details about the time points are provided in Fig. S2).

**TABLE 2** Effect of antimicrobial agent applications on antimicrobial-resistant gene prevalence in soil and tomato leaves between TP7 and TP9[a]

| AMR class | Leaf | | | | Soil | | | |
|---|---|---|---|---|---|---|---|---|
| | Ct | Cu | St | Tr | Ct | Cu | St | Tr |
| Aminoglycoside-resistance (5) | 43 | 64 | 45 | 67 | 100 | 100 | 100 | 100 |
| Beta-lactam resistance (1) | 0 | 18 | 9 | 8 | 0 | 0 | 17 | 0 |
| Class A beta-lactamase (22) | 0 | 36 | 27 | 0 | 38 | 8 | 0 | 17 |
| Class B beta-lactamase (9) | 29 | 18 | 0 | 17 | 100 | 92 | 100 | 100 |
| Class C beta-lactamase (11) | 100 | 73 | 73 | 58 | 100 | 92 | 100 | 100 |
| Class D beta-lactamase (13) | 71 | 36 | 27 | 50 | 100 | 100 | 100 | 100 |
| Erythromycin resistance (1) | 14 | 0 | 0 | 0 | 25 | 25 | 8 | 8 |
| Fluoroquinolone resistance (11) | 14 | 27 | 36 | 17 | 38 | 58 | 83 | 50 |
| IgG binding protein A precursor (1) | 0 | 0 | 0 | 0 | 0 | 0 | 0 | 0 |
| Macrolide lincosamide streptogramin_b (1) | 100 | 100 | 82 | 83 | 100 | 100 | 100 | 92 |
| Multidrug resistance efflux pump (2) | 14 | 0 | 0 | 8 | 0 | 8 | 0 | 0 |
| Panton-Valentine leukocidin chain F (1) | 0 | 0 | 0 | 0 | 0 | 0 | 0 | 0 |
| *Staphylococcus aureus* (1) | 57 | 45 | 45 | 50 | 25 | 0 | 8 | 8 |
| Tetracycline efflux pump (2) | 57 | 55 | 64 | 50 | 25 | 25 | 58 | 58 |
| Vancomycin resistance (2) | 0 | 9 | 0 | 0 | 0 | 0 | 0 | 0 |
| Cumulative score | 500 | 482 | 409 | 408 | 650 | 608 | 675 | 633 |

[a]Number in parentheses refers to the number of ARGs studied; ARG: antimicrobial resistant gene; Cu, St, and Tr: plots to which copper hydroxide, streptomycin, or propiconazole were applied, respectively; C and Ct: non-treated control rows and plot, respectively. Cumulative score: sum of the prevalence for the given treatment group (max value = 1,500). $N = 12$ samples per group treated with one antimicrobial agent. $N = 8$ samples for the non-treated plots. TP6: before the first application of the antimicrobials; TP7–9: after the application of antimicrobials on a weekly basis (additional details about the time points are provided in Fig. S2).

During the growing season (TP6–TP9), soil ARG abundance was stable (Fig. 4), but diversity increased from 18 ARGs at TP6 to 29–30 at TP7–TP9 (Fig. S8C). Leaf ARGs similarly increased (from 10 at TP6 to 20–32 at TP7–TP9; Fig. S8C). At TP9, the GM group had significantly more ARGs in soil (18.0 [15.5–20.5]) than others (<12.7 [8.9–16.5]), and St-treated soils had more ARGs (17.0 [14.5–19.5]) than Cu (10.0 [3.4–16.7]) or Tr (11.0 [2.4–19.6]; $P < 0.05$; Fig. S8C). Manure + Cu/Tr plots had the fewest ARGs, whereas untreated plots harbored the most (Fig. S8C).

Leaf ARG composition was more variable than soil (Fig. 4). ARG counts peaked at TP7 (9.1 [7.5–10.6]), then decreased at TP8 and TP9. Fewer ARGs overlapped between glyphosate/manure and AM-treated plots (32.6% and 30.4%, respectively; Fig. S8B). At TP7, M, G, and GM plots had ≥1.7× more ARGs in leaves than C group (5.75 [0.1–11.5]; $P < 0.05$; Fig. S8C). Untreated plots consistently showed the lowest ARG levels in leaves. Despite overlapping ARGs in soil and leaves (TP7–TP9), soil ARG diversity partially explained leaf ARG composition ($r^2 = 0.27$; $P < 0.01$). Co-occurrence patterns varied by treatment, with stronger correlations in M, GM, and Ct groups ($r^2 = 0.31–0.36$) than in C, G, or AM-treated groups ($r^2 = 0.20–0.29$).

## Effects of agricultural practices on soil and leaf microbiomes

Soil microbiome at TP1 was dominated by Proteobacteria (38%), Bacteroidetes (18%), Acidobacteria (10%), and others. Manure (TP2) had a distinct microbiome but did not significantly alter soil composition by TP3 vs control. Temporal shifts were more notable, with increased Acidobacteria and reduced Cyanobacteria at TP3. Glyphosate significantly reduced soil microbial diversity (Faith PD and Shannon indices) at TP4, particularly when combined with manure ($P < 0.01$; Table 3).

Distinct microbial communities were observed between roots and leaves of transplanted seedlings at TP5. Roots were rich in Proteobacteria (44%) and Bacteroidetes (24%), while leaves were mostly Cyanobacteria (90%). Leaf microbiomes remained stable from TP5 to TP6; soil microbiomes also stabilized after TP4.

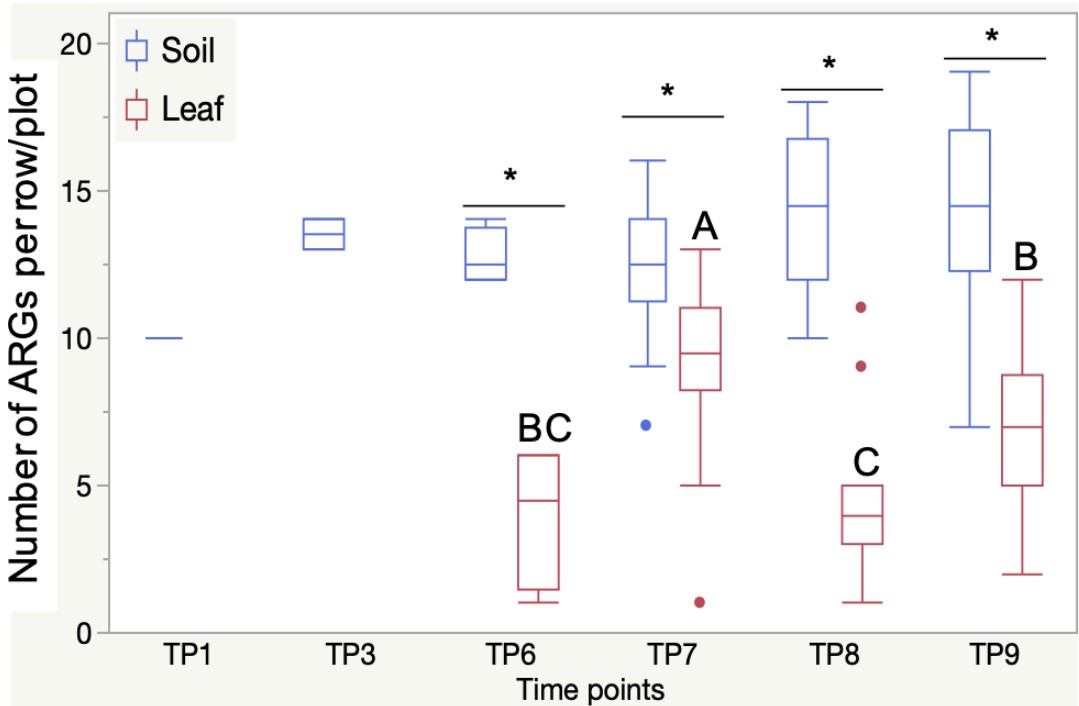

FIG 4 ARG abundance in soil and leaf collected between TP1 and TP9. Impact of health management practices and disease management practices on the AMR gene profiles in the soil collected between TP6 and TP9. Stars indicate significant differences in the abundance of AMR detected between the soil and the leaf samples for a given time point ($P < 0.05$). Letters (A–C) indicate significant differences in the abundance of AMR detected in the leaf samples between time points ($P < 0.05$).

A significant shift occurred in the soil microbiome between TP6 (pre-antimicrobial treatment) and TP7 (post-antimicrobial treatment). TP7 soils had reduced Bacteroidetes (e.g., Dyadobacter, Flavobacterium, Sphingobacterium, *Taibaiella,* and *Chitinophaga* sp. BN130233) and increased Firmicutes (especially, *Turicibacter* and *Romboutsia*), Chloroflexi (e.g., HSB OF53-F07, FCPS473, and JG30a-KF-32), Planctomycetes (e.g., *Pirellulaceae* and Pirellula), and Actinobacteria (e.g., *Acidothermus, Cellulomonas, Conexibacter,* and *Nocardioides*) compared to TP6 $P < 0.01$).

Soil samples consistently showed greater microbial diversity (Faith's PD, Shannon index; ~37×) than leaves (Table 3). Across TP6–TP9, manure-treated plots had the highest microbial diversity, while glyphosate- and Cu-treated plots had the lowest in both soil and leaves ($P < 0.01$). Tr-treated plots also had lower diversity in leaves than controls. Additional details about the microbial diversity and taxonomic differences are provided in Table S4. The soil microbiome was principally composed of Proteobacteria (28%–47%), Acidobacteria (9%–25%), Bacteroidetes (10%–23%), Verrucomicrobia (0%–10%), Firmicutes (0.6%–16%), Cyanobacteria (0%–13%), and Chloroflexi (0%–8%; Fig. 5D). On the other hand, the leaf microbiome was principally composed of Cyanobacteria (0%–100%) and Proteobacteria (0%–29%; Fig. 5E) across all time points.

## Correlations between the microbiome and ARG profiles

Out of 760 ASVs (amplicon sequence variants; species-level), 144 (19%) from soil and tomato leaf microbiomes were positively correlated with ARG abundance (36/87 genes; $0.28 < r^2 < 0.52$; $P < 0.05$). Correlation patterns differed by sample type, with only five ASVs shared across both (e.g., *Acidobacteria* SCN 69-37, *Ellin6529, Luteimonas composti, Sphingomonas jaspsi*), all of which were low in abundance (<1%). However, most correlated ARGs (34/36) were shared between soil and leaf microbiomes.

**TABLE 3** Alpha diversity (Faith's PD and Shannon indices) in soil and tomato leaves between agricultural practices over time[a]

| Sample type | TP | Alpha diversity of tomato plots with glyphosate and dairy manure application | | | | | | | | | |
|---|---|---|---|---|---|---|---|---|---|---|---|
| | | FaithPD | | | | | Shannon | | | | |
| | | GAR | C | G | GM | M | GAR | C | G | GM | M |
| Soil | TP3 | [81–93] | NS | | | NS | [6.24–6.5] | NS | | | NS |
| Soil | TP4 | [72–90] | A | B | C | AB | [5.95–6.44] | A | B | C | AB |
| Soil | TP6 | [79–95] | B | B | AB | A | [6.19–6.54] | B | B | AB | A |
| Soil | TP7 | [75–96] | BC | C | AB | A | [6.07–6.56] | B | B | AB | A |
| Soil | TP8 | [74–94] | NS | NS | NS | NS | [6.03–6.51] | NS | NS | NS | NS |
| Soil | TP9 | [72–96] | A | B | B | C | [6.02-6.56] | A | B | B | C |
| Leaf | TP6 | [7–23] | C | B | A | | [2.14–2.9] | C | B | A | |
| Leaf | TP7 | [4–31] | A | B | B | AB | [1.18–3.29] | A | C | B | A |
| Leaf | TP8 | [3–21] | B | C | D | A | [1.01–2.44] | B | C | C | A |
| Leaf | TP9 | [4–28] | B | C | A | A | [1.14–3.7] | C | B | A | A |
| Sample type | TP | Alpha diversity of tomato plots with copper, streptomycin, or triazole applications | | | | | | | | | |
| | | FaithPD | | | | | Shannon | | | | |
| | | GAR | Ct | Cu | St | Tr | GAR | Ct | Cu | St | Tr |
| Soil | TP7 | [75–96] | A | B | B | A | [6.07–6.56] | A | B | B | A |
| Soil | TP8 | [74–94] | A | B | A | A | [6.03–6.51] | A | B | A | A |
| Soil | TP9 | [72–96] | NS | NS | NS | NS | [6.02–6.56] | NS | NS | NS | NS |
| Leaf | TP7 | [4–31] | A | C | AB | B | [1.18–3.29] | A | C | ABC | B |
| Leaf | TP8 | [3–21] | NS | NS | NS | NS | [1.01–2.44] | A | AB | AB | B |
| Leaf | TP9 | [4–28] | NS | NS | NS | NS | [1.14–3.7] | B | AB | B | A |

[a]TP: time point; G, M, and GM: plots applied with glyphosate, dairy manure, or both, respectively; Cu, St, and Tr: plots copper hydroxide, streptomycin, or propiconazole applications, respectively; C and Ct: non-treated control rows and plot, respectively. Letters (A–D) indicate statistical groups ($P < 0.01$). GAR: minimum and maximum index values, respectively, recorded for the designated groups and time points. HMP and DMP: health and disease management practices. NS: no significant difference detected ($P > 0.01$). TP3: before glyphosate application; TP4: after glyphosate application; TP6: before the first application of the antimicrobials; TP7–9: after the application of antimicrobials on a weekly basis.

In soil, 112 ASVs positively correlated with 34 ARGs linked to aminoglycosides ($n = 65$ ASVs; especially *aacC1*, *aacC2*), fluoroquinolones ($n = 57$ ASVs; especially *QnrB-5*, *AAC(6)-Ib-cr*), class A ($n = 40$ ASVs; especially *SFO-1*) and class C ($n = 43$ ASVs; especially *LAT*) beta-lactamases, macrolides ($n = 33$ ASVs; *ermC*), and erythromycin ($n = 30$ ASVs; especially *ereB*; Table 4). Most correlated soil ASVs belonged to *Proteobacteria* ($n = 39$ ASVs; especially *Alphaproteobacteria*, *Gammaproteobacteria*), *Bacteroidetes* ($n = 24$ ASVs; especially *Chitinophagaceae*), and *Actinobacteria* ($n = 12$ ASVs; subgroups 1 and 6). Notably, ASVs associated with aminoglycoside resistance were inversely correlated with those linked to fluoroquinolone resistance ($r^2 = -0.83$; $P < 0.01$). Additionally, 120 soil ASVs were negatively correlated with ARGs (Table S5).

In tomato leaves, 26 ASVs were positively correlated with 14 ARGs, mainly related to macrolides ($n = 23$ ASVs; especially *ermC*, *mrsA*), class C ($n = 23$ ASVs; especially *ACT-1*, *MIR*), and class D (23 ASVs; especially *OXA-58/60*) beta-lactamases (Table 4). These ASVs were primarily from *Proteobacteria* ($n = 13$ ASVs; especially *Alphaproteobacteria*, *Sphingomonadaceae*), *Bacteroidetes* ($n = 9$ ASVs; *Hymenobacteraceae*), and *Acidobacteria* ($n = 7$ ASVs). No ASVs were negatively correlated with ARGs in leaves.

## Antimicrobial residues in soil

LC-MS/MS analysis showed no detectable streptomycin before application. Residues were detected at 0.033 ± 0.004 ppm at TP7 (6 days after first AM application), and 0.019 ± 0.007 ppm at TP8 (after the fourth application), with no significant differences across treatment types (C, G, M, GM; Fig. S9A). A significant 8.6 ± 3.4-fold increase occurred by TP9 (0.15 ± 0.03 ppm; $P < 0.01$), 6 days after the fifth application. Interestingly, untreated plots (C) had higher streptomycin residues at TP9 (0.19 ± 0.02 ppm) than manure-treated plots (M; 0.13 ± 0.02 ppm; $P < 0.01$; Fig. S9A).

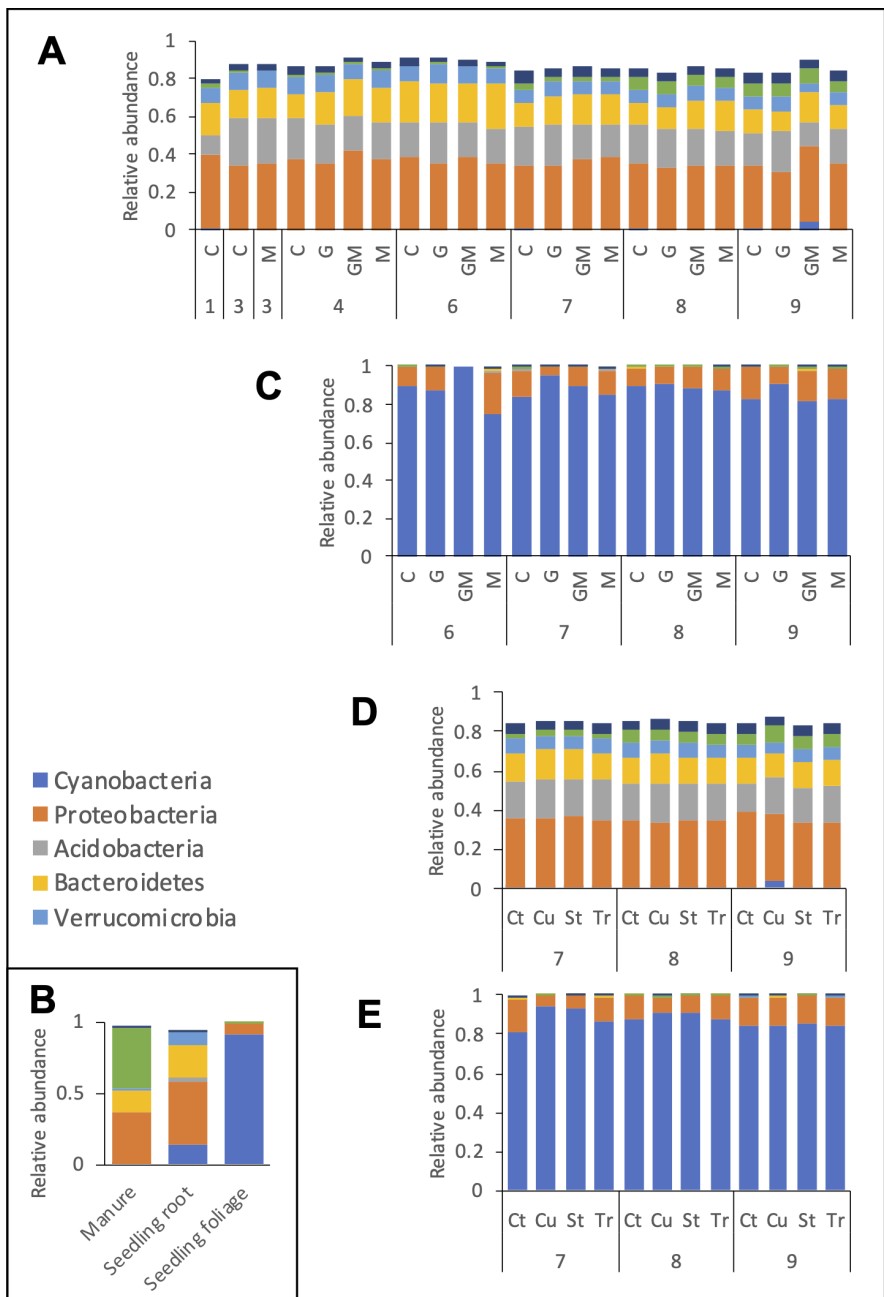

**FIG 5** Impacts of agricultural practices on the soil and tomato leaf microbiomes at the phylum level. Microbiome composition at the phylum level of the soil (A and D) and tomato leaf tissues (C and E) over time (time point TP1–TP9). Panels A and D indicate the impact of manure and glyphosate on soil and tomato leaf microbiomes over time, respectively. Panels D and E indicate the impact of pesticides (copper, streptomycin, and propiconazole) on soil and tomato leaf microbiomes over time, respectively. (B) Microbiome composition at the phylum level of the agricultural inputs (pure dairy manure collected at TP2 and seedlings collected at TP5). G, GM, and M: plots to which glyphosate and/or manure were applied; (C): non-treated plots. Cu, St, and Tr: plots to which copper hydroxide, streptomycin, or propiconazole were applied, respectively; C and Ct: non-treated control rows and plot, respectively.

Copper residues ranged from 1.92 ± 0.57 to 3.3 ± 1.14 mg/kg across all plots (including non-treated) from TP1 to TP9 (Fig. S9B). While copper-treated plots had consistently higher concentrations than untreated ones, levels remained stable over time without cumulative buildup.

**TABLE 4** Positive correlation between the tomato microbiome and associated ARGs[a]

| Sample type | AMR class | ARG | ASV (genus/species level) |
|---|---|---|---|
| Soil | Class C and D beta-lactamase, fluoroquinolone, and macrolide | LAT, OXA-60, AAC(6)-lb-cr, and ermC | Aridibacter famidurans |
| | | | Candidate division TM7 bacterium LY2 |
| | | | Dokdonella ginsengisoli |
| | | | Pseudomonas sp. MYb218 |
| | | | Saccharum hybrid cultivar |
| | | | Soil bacterium WF55 |
| | | | Solitalea canadensis |
| | | | Sphingosinicella sp. |
| | | | Uncultured bacterium KF-JG30-18 |
| | | | Uncultured bacterium SJA-5 |
| | | | Uncultured eubacterium WD247 |
| | | | Uncultured lamia sp. |
| | | | Uncultured Prosthecomicrobium sp. |
| | | | Unidentified archaeon SCA1173 |
| | | | Verrucomicrobia bacterium SCGC AG-212-E04 |
| | | | Zavarzinella formosa |
| | Aminoglucoside, class D beta-lactamase, and macrolide | aacC2, OXA-60, ermA B and C, and mefA | Bacterium Ellin517 |
| | | | Bacterium MI-37 |
| | | | Niastella yeongjuensis |
| | Aminoglucoside and fluoroquinolone | aacC1 and 2 and AAC(6)-lb-cr | Candidatus Uhrbacteria bacterium RIFOXYC2_FULL_47 |
| | | | Gemmatimonadetes bacterium Ellin7146 |
| | | | Kofleria flava |
| | | | Uncultured Ohtaekwangia sp. |
| | | | Uncultured sludge bacterium A9 |
| | Aminoglucoside | aacC1 and 2 | Anaeromyxobacter dehalogenans |
| | | | Delta proteobacterium WX152 |
| | | | Dongia mobilis |
| | | | Luteimonas composti |
| | Aminoglucoside, class C beta-lactamase, macrolide, and tetracycline | aacC1, LAT, ermA, and tetA | Acidobacteria bacterium WX184 |
| | | | Acinetobacter sp. IMCC12751 |
| | | | Aquicella siphonis |
| | | | Bacterium enrichment culture clone BBMC-4 |
| | | | Nostocoida limicola III |
| | | | Planctomycetales bacterium Ellin6207 |
| | | | Uncultured Acidobacterium sp. |
| | | | Uncultured bacterium 16H1 |
| | | | Uncultured Dyadobacter sp. |
| | | | Uncultured Rudaea sp. |
| | | | Uncultured Sporichthya sp. |
| | Aminoglycoside and tetracycline | aacC1 and 2, aadA1, tetA | Alpha proteobacterium U9-1i |
| | | | Chlorobi bacterium OLB5 |
| | | | Oryzihumus leptocrescens |
| | | | Uncultured sludge bacterium H8 |
| | Class D beta-lactamase, erythromycin, and fluoroquinolone | OXA-60, ereB, and AAC(6)-lb-cr | Chitinophaga niastensis |
| | | | Chitinophaga pinensis DSM 2588 |

(*Continued on next page*)

**TABLE 4** Positive correlation between the tomato microbiome and associated ARGs[a] (*Continued*)

| Sample type | AMR class | ARG | ASV (genus/species level) |
|---|---|---|---|
| | | | Flavihumibacter petaseus NBRC 106054 |
| | | | Uncultured Alicyclobacillus sp. |
| | | | Uncultured Longilinea sp. |
| | Aminoglycoside and erythromycin | aacC1 and 2 and ereB | Acidobacteriaceae bacterium KBS 83 |
| | | | Aquimonas sp. enrichment culture clone 03SUJ1 |
| | | | Bdellovibrio sp. ETB |
| | | | Beta proteobacterium WF17 |
| | | | Candidate division TM7 bacterium JGI 0001002-L20 |
| | | | Dubosiella newyorkensis |
| | | | Gemmatimonadetes bacterium SCN 70-22 |
| | | | Holophaga sp. WY42 |
| | | | Oligoflexus tunisiensis |
| | | | Paenibacillus anaericanus |
| | | | Reyranella sp. |
| | | | Runella sp. NBRC 15147 |
| | | | Saccharomonospora viridis DSM 43017 |
| | | | Sphingomonas jaspsi |
| | | | Taibaiella sp. |
| | | | Uncultured Chloroflexus sp. |
| | | | Uncultured eubacterium WD244 |
| | | | Uncultured eubacterium WD264 |
| | | | Uncultured rape rhizosphere bacterium wr0025 |
| | | | Uncultured Steroidobacter sp. |
| | Class B and D beta-lactamase, erythromycin, and fluoroquinolone | IMP-2 group, MOX, ereB, and QnrB-5 and 8 group | Alkanindiges illinoisensis |
| | | | Delta proteobacterium LX33 |
| | | | Filamentous bacterium Plant1 Iso8 |
| | Class A beta-lactamase and fluoroquinolone | SFO-1 and QnrB-5 group | Acidobacteria bacterium SCN 69-37 |
| | | | Arenimonas daechungensis |
| | | | Bacterium Ellin507 |
| | | | Bacterium Ellin5290 |
| | | | Bacterium Ellin6529 |
| | | | Bacterium enrichment culture clone auto195_4W |
| | | | Bacterium enrichment culture clone D8 |
| | | | Bdellovibrionales bacterium Ga0074137 |
| | | | Candidatus Yanofskybacteria bacterium GW2011_GWF2_43_596 |
| | | | Cyanobacteria/Melainabacteria group bacterium S15B-MN24 CBMW_12 |
| | | | Cytophaga aurantiaca DSM 3654 |
| | | | Ephemera danica |
| | | | Flavisolibacter ginsengisoli |
| | | | Granulicella paludicola |
| | | | Hyaloperonospora arabidopsidis |
| | | | Isosphaera sp. |
| | | | Malawimonas jakobiformis |
| | | | Methylovorus sp. MM1 |
| | | | Moheibacter sediminis |
| | | | Rhodobacter sp. 7B-409 |

(*Continued on next page*)

**TABLE 4** Positive correlation between the tomato microbiome and associated ARGs[a] (*Continued*)

| Sample type | AMR class | ARG | ASV (genus/species level) |
|---|---|---|---|
| | | | Spumella-like flagellate JBM08 |
| | | | Uncultured Anaeromyxobacter sp. |
| | | | Uncultured bacterium DA023 |
| | | | Uncultured bacterium mle1-25 |
| | | | Uncultured Bacteroides sp. |
| | | | Uncultured Haliangium sp. |
| | | | Uncultured Opitutus sp. |
| | | | Uncultured Pseudolabrys sp. |
| | | | Uncultured Stella sp. |
| | Aminoglycoside and class A, B, and C beta-lactamase | aphA6, SFO-1, SHV(156G), IMP-2-5 and -12 group, DHA | Algoriphagus terrigena DSM 22685 |
| | | | Bacteroidetes bacterium UKL13-3 |
| | | | Gemmatimonadetes bacterium WY71 |
| | | | Terrimonas sp. 16-45A |
| | | | Uncultured eubacterium WD2123 |
| | | | Uncultured Ferruginibacter sp. |
| | Aminoglycoside and class B and C beta-lactamase | aphA6, IMP-2-5 and -12 group, DHA | Bacterium enrichment culture clone auto9_4W |
| | | | Chitinophaga sp. BN130233 |
| | | | Cytophaga hutchinsonii ATCC 33406 |
| | | | Uncultured Cellvibrio sp. |
| | | | Uncultured Pedosphaera sp. |
| | | | Uncultured verrucomicrobium DEV009 |
| Leaf | Beta-lactamase | mecA | Uncultured eubacterium WD260 |
| | | | Actinobacteria bacterium IMCC26207 |
| | | | Uncultured Blastococcus sp. |
| | | | Mycoplasma sp. A4 |
| | | | Bacterium Ellin6543 |
| | | | Uncultured Ruminiclostridium sp. |
| | | | Uncultured Singulisphaera sp. |
| | | | Alpha proteobacterium SK200a-2 |
| | | | Pythium ultimum |
| | | | Sorangium cellulosum |
| | | | Candidatus Xiphinematobacter americani |
| | Class C beta-lactamase | ACT-1 group and MIR | Sphingomonas jaspsi |
| | | | Uncultured Holophaga sp. |
| | Aminoglycoside and class C beta-lactamase | aphA6, ACT-1 group, and MIR | Antarctic bacterium GA028 |
| | | | Gemmatimonadaceae bacterium LWQ133 |
| | | | Hymenobacter algoricola |
| | Aminoglycoside and macrolide | aadA1 and msrA | Nannocystis sp. 0558-ZXW50 |
| | | | Uncultured Acidobacterium UA3 |
| | Macrolide | ermC | Sphingomonas sp. LP28 |
| | | | Acidobacteria bacterium SCN 69-37 |
| | | | Bacterium enrichment culture clone D8 |
| | | | Hymenobacter sp. DG25A |
| | | | Lactobacillus vespulae |
| | | | Luteimonas composti |
| | | | Mucilaginibacter sp. HME6834 |
| | | | Pontibacter sp. LX8 |
| | | | Uncultured bacterium 5G12 |
| | Class D beta-lactamase and macrolide | OXA-58 and msrA | Azolla filiculoides |
| | | | Nitrospira japonica |

**TABLE 4** Positive correlation between the tomato microbiome and associated ARGs[a] (*Continued*)

| Sample type | AMR class | ARG | ASV (genus/species level) |
|---|---|---|---|
| | | | Tectona grandis |
| | | | Clostridium sp. CL-2 |
| | | | Buchnera aphidicola (Anoecia fulviabdominalis) |
| | | | Bacterium Ellin6529 |
| | | | Bacterium LWQ8 |
| | | | Geodermatophilus sp. |
| | | | Sphingoaurantiacus polygranulatus |

[a]Only microbiome-ARG correlations with a *P*-value under 0.01 are displayed in the table.

## Repeated antimicrobial applications did not alter the physico-chemical properties of soil

The analysis of the physico-chemical properties of the soil collected at TP1 confirmed that the field was homogeneous (Fig. S10). The agricultural practices had no distinct effect on the physico-chemical properties of the soil. However, differences in soil properties were observed over time. Most of the differences in soil properties were observed between TP1 and TP7. TP7 soil was characterized by a neutral pH, high levels of phosphorus, sodium, potassium, organic matter, and free nitrogen, while TP1 soil was characterized by a lower pH and higher levels of calcium, aluminum, and magnesium. Interestingly, the physico-chemical properties of the soil after TP7 shifted toward the initial properties observed in TP1.

## DISCUSSION

The contribution of plant agricultural practices to the burden of AMR is increasingly recognized (1, 14, 45). Despite the widespread use of pesticides in plant agriculture to ensure crop yield and quality (12, 46), their impact on microbial communities and resistance development remains underexplored. Our study underscores the need to assess how intensive agricultural inputs influence AMR dissemination. Resistant bacteria and genes arising in crop production environments can be transferred through the food chain, irrigation water, or aerosols, ultimately threatening human health by limiting treatment options and increasing the risk of foodborne outbreaks linked to resistant pathogens (47).

Our study, along with several others, demonstrated that both animal manure (e.g., squeezed dairy manure) and repeated applications of agricultural pesticides (e.g., glyphosate, copper hydroxide, and streptomycin) significantly altered AMR patterns in the soil and phyllosphere microbiomes of processing tomato plants (1, 14, 25, 45, 48). Notably, manure application increased the abundance of copper-, streptomycin-, and beta-lactam-resistant bacteria, particularly during the growing season (June–July 2019, TP6–TP9), indicating environmental factors may influence CARB persistence. Manure-treated plots also exhibited the highest prevalence of soil ARGs and class C beta-lactamase genes (e.g., ACT-1, LAT, MIR) on tomato leaves during this period. Although we did not assess whether these ARGs were plasmid- or chromosome-encoded, our findings support previous concerns regarding manure as a reservoir for clinically relevant AMR bacteria, including AmpC-producing enteric bacteria (11, 49–51). Alternatives such as application of inorganic fertilizer or pre-treated manure (composting rather than heat-shock treatment) may offer safer options by reducing the dissemination of AMR in the environment (51–54). In addition to alternative manure management strategies, the use of emerging formulations and application strategies of antimicrobials to control plant diseases has shown promise as sustainable alternatives to conventional antimicrobials. For example, the use of silver, copper oxide, and zinc oxide nanoparticles has been reported to effectively suppress plant and soilborne bacterial pathogens, while reducing the risk of conventional antimicrobial overuse (55–60). Biological control agents (BCAs)

are another alternative to conventional antimicrobials, and in recent years, new BCAs with improved efficacy as well as plant growth promotion are being explored (61). Of particular interest is bacteriophage-mediated biocontrol (62, 63). Bacteriophages are naturally occurring in the environment and have the generally recognized as safe status from the US Food and Drug Administration (64). They have been successfully used in agriculture to manage destructive diseases such as bacterial spot and speck of tomato and pepper, citrus canker, and Pierce's disease of grapevine. Lastly, the advancement of intelligent or smart spray technologies has dramatically improved the sustainability of antimicrobials through the reduction in chemical use and off-target run-off and drift, and slowing the development of AMR (65, 66). Incorporating these alternatives into integrated pest and resistance management frameworks can mitigate the spread of AMR and ultimately improve plant and public health.

The postemergence application of glyphosate led to a transient increase in CARB abundance in soil, detectable four days after treatment. However, CARB levels declined later in the growing season. Conversely, glyphosate-treated plots consistently showed reduced CARB and ARG prevalence on tomato leaves, particularly for copper, streptomycin, and beta-lactam resistance, suggesting a potential antagonistic effect on phyllosphere bacteria. These findings align with previous studies showing glyphosate's antimicrobial activity and its potential to disrupt microbial communities (13, 14, 67). Moreover, our data support the hypothesis that glyphosate-induced shifts in the soil microbiome may influence microbial transfer to plant tissues. Interestingly, the suppressive effects of glyphosate on leaf CARB and ARGs were nullified when combined with manure. Plots treated with both inputs showed the highest prevalence of soil ARGs and aminoglycoside resistance genes (e.g., *aadA1*, *aphA6*) in leaves. This interaction underscores the dominant role manure plays in shaping resistance profiles and highlights the potential risk of co-applying manure with herbicides in agricultural systems. These findings are agronomically relevant, but also critical from a food safety perspective, as they may create favorable conditions for resistant bacteria to colonize edible plant surfaces, increasing the risk of human exposure through consumption of raw or minimally processed produce.

Streptomycin and propiconazole treatments were associated with the lowest ARG prevalence on tomato leaves, particularly for class B (IMP-12), C (ACT-1, DHA, MIR, MOX), and D (OXA-58, OXA-60) beta-lactamases, erythromycin (*ereB*), and multidrug efflux pumps (*oprJ*, *oprM*), compared to untreated plots. These results suggest that azole fungicides, like propiconazole, can impact phyllosphere resistomes similarly to antibacterial agents. Given previous concerns about azole-driven AMR evolution in fungi (24), our findings highlight the need for further research into their broader effects on bacterial resistance. Of note, the combined application of glyphosate and propiconazole—especially when paired with manure—appeared to enhance the suppressive effect on CARB in the phyllosphere. This combinatorial interaction warrants further investigation, particularly into the potential of glyphosate to modulate the antimicrobial effects of other agrichemicals. From a public health standpoint, understanding these interactions is essential because fungicide- and herbicide-driven resistance mechanisms may overlap with clinical resistance determinants, potentially narrowing therapeutic options against foodborne and opportunistic pathogens.

We acknowledge that our study is based on a single growing season (March–July 2019) and that data collection ended early due to severe flooding in August 2019. Post-flooding, evidence of CARB dispersal across plots and uncontrolled environmental mixing led us to terminate the experiment to preserve data integrity. Consequently, we could not assess treatment impacts at harvest. Nonetheless, our findings are consistent with previous studies, including similar manure-associated AMR trends observed in China (25), reinforcing the relevance of our conclusions.

## Conclusion

Our study demonstrates that while individual agricultural practices may exert limited influence on the AMR composition in processing tomato systems, their combination can significantly alter both the abundance and diversity of CARB and ARGs. Notably, herbicides (glyphosate) and fungicides (propiconazole) had measurable impacts on the phyllosphere resistome, comparable to those of traditional antimicrobial agents such as copper hydroxide and streptomycin. This study aligns with the One Health framework, which emphasizes the interconnectedness of human, animal, and environmental health. Resistant bacteria and resistance genes originating in crop production systems do not remain confined to the agricultural environment but can disseminate through food products, water systems, and soil–human contact interfaces. This highlights the urgent need for integrated mitigation strategies that consider agricultural inputs, pathogen control innovations, and waste management practices in order to protect plant health, food safety, and public health on a global scale. Furthermore, these findings highlight the need for further investigation into the effects of diverse agricultural inputs on microbial communities and resistance dynamics. Similar studies across a broader range of crop systems will be essential to understand and mitigate emerging AMR risks in plant agriculture. Moreover, comprehensive characterization of the impacts of agricultural management on plant-associated microbiomes can inform the development of more sustainable and safer farming practices. Ultimately, deeper insight into how agricultural practices shape microbial communities may support targeted microbiome manipulation and foster the discovery of effective, non-biocide-based alternatives to conventional pesticide use.

## ACKNOWLEDGMENTS

We thank Rachel Kaufman, Dr. Dipak Kathayat, Carlos Saint-Preux, Margaret Moodispaw, Dr. Veerupaxagouda Patil, Dr. Yosra A. Helmy, Dr. Vishal Srivastava, and Dhawni Parsana, for their technical support.

This research was supported by the Centers for Disease Control (CDC) contract # 200-2018-02920, and by state and federal funds appropriated to The Ohio State University College of Food, Agricultural, and Environmental Sciences.

## AUTHOR AFFILIATIONS

[1]Department of Animal Sciences, The Ohio State University, Wooster, Ohio, USA
[2]Department of Plant Pathology, The Ohio State University, Wooster, Ohio, USA
[3]University of Illinois, Urbana-Champaign, Illinois, USA

## AUTHOR ORCIDs

Loic Deblais  http://orcid.org/0000-0002-6290-3956
Alejandra M. Jimenez Madrid  http://orcid.org/0000-0001-8082-2124
Melanie L. Lewis Ivey  https://orcid.org/0000-0001-7520-9026
Gireesh Rajashekara  http://orcid.org/0000-0003-2443-6733

## FUNDING

| Funder | Grant(s) | Author(s) |
| --- | --- | --- |
| Centers for Disease Control and Prevention | 200-2018-02920 | Gireesh Rajashekara |

## DATA AVAILABILITY

Illumina reads are available for all the field samples sequenced in this study under NCBI Bioproject no. PRJNA1284401, under Sequence Read Archive (SRA) SRR34317913.

## ADDITIONAL FILES

The following material is available online.

### Supplemental Material

**Supplemental figures (Spectrum02003-25-s0001.pdf).** Figures S1, S2, and S4 to S10.

**Figure S3 (Spectrum02003-25-s0002.pdf).** Impact of glyphosate and manure on the levels of culturable antibiotic resistant bacteria in the soil and leaf samples collected between TP4 and TP9.

**Table S1 (Spectrum02003-25-s0003.docx).** Agricultural practices and samples' collection.

**Table S2 (Spectrum02003-25-s0004.docx).** Details about the Microbial DNA qPCR Array used for the detection of antibiotic resistance genes.

**Table S3 (Spectrum02003-25-s0005.docx).** Antimicrobial resistant gene profile from the soil collected on TP1 and the manure applied on the field at TP2.

**Table S4 (Spectrum02003-25-s0006.docx).** Microbiome composition differences detected between the agricultural practices at the genus/species levels.

**Table S5 (Spectrum02003-25-s0007.docx).** Negative correlations in soil samples between the agricultural microbiome and associated antimicrobial resistance gene profiles.

### Open Peer Review

**PEER REVIEW HISTORY (review-history.pdf).** An accounting of the reviewer comments and feedback.

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
