## [Reviewer comments · Microbiology Spectrum]

Microbiology Spectrum

Dairy manure, glyphosate, and antimicrobials (copper, streptomycin and triazole) modulated the composition of antimicrobial resistance at the gene and microbial levels in a processing tomato field.

Gireesh Rajashekara, Loic Deblais, Gabrielle Derippe, Madeline Horvat, Sochina Ranjit, Vincent Moulia, Alejandra Jimenez Madrid, Michael Kauffman, Francesca Rotondo, Melanie Ivey, and Sally Miller

Corresponding Author(s): Gireesh Rajashekara, University of Illinois Urbana-Champaign

Review Timeline:

Submission Date:	June 30, 2025
Editorial Decision:	September 5, 2025
Revision Received:	November 4, 2025
Accepted:	December 10, 2025

Editor: Katharina Kujala

Reviewer(s): Disclosure of reviewer identity is with reference to reviewer comments included in decision letter(s). The following individuals involved in review of your submission have agreed to reveal their identity: Muhammad Afzal (Reviewer #2)

Transaction Report:

DOI: <https://doi.org/10.1128/spectrum.02003-25>

Re: Spectrum02003-25 (**Dairy manure, glyphosate, and antimicrobials (copper, streptomycin and triazole) modulated the composition of antimicrobial resistance at the gene and microbial levels in a processing tomato field.**)

Dear Prof. Gireesh Rajashekara:

Thank you for the privilege of reviewing your work. Below you will find my comments, instructions from the Spectrum editorial office, and the reviewer comments.

Revision Guidelines

Sincerely,
Katharina Kujala
Editor
Microbiology Spectrum

Reviewer #2 (Comments for the Author):

General Comments

I have read the paper. Overall, the article is well structure and written. However, the topic is not novel several studies have been published on RGs. Furthermore, research gap needs to discuss in the introduction section more explicitly. How it advances our

understanding of agriculture practices such as manure application and use of pesticides to influence AMR? In the introduction section, relevance of the research question to public health and food safety could be focused in detail. My further comments and suggestions are given below.

General comments;

Several studies used nano-particles (NPs) (<https://doi.org/10.1016/j.stress.2025.100917>; <https://doi.org/10.1016/j.plana.2024.100080>; <https://doi.org/10.3390/plants12132461>) for different bacterial pathogens control. Discuss this aspect in this article related to AMR.

In the introduction, clearly explain research gap, significance, relevance to human health, and significance of AMR study in tomato farming.

Recent studies needed to include particularly related to the combine effects of herbicides, pesticides, and manure on AMR in context of their impact on soil microbial communities in the Agri-ecosystems.

Microbial community shift, must be explain in context with AGRs in soil and plant leaves.

In the results section p values in the figures and tables must be included consistently. Detail captions needed.

Discussion section needs deep literature in term of public health, food safety.

References style is inconsistent.

Specific comments

Abstract: Abstract need more streamline for quick reading, for example, summarize AGRs mention in this section without excessive detail.

M & M: Chemical dosages and application methods are not explained in sufficient detail. Provide concentration of chemical (glyphosate) and manure applied, compared to typical agricultural practices. In addition, why such concentration has been selected?

The study mentioned 64 experimental plots, but plots selection and randomization are not clear. How much replicates? How the treatments were assigned to each plot?

Why ANOVA and Tukey tests for statistical analysis were used while alternative tests are available. Microbial data processing and analysis need to explain.

Results: Explain microbial diversity indices in more detail treatment wise.

What are the biological mechanisms to shift the said microbial community?

Language and Grammar: The language is clear and technical. However, simplified sentences for better clarity. For instant, the sentence "Glyphosate treatment, alone or with manure, significantly increased total soil CARB by TP4" could be rephrased as "Glyphosate treatment, with or without manure, significantly increased CARB levels in the soil by TP4." Additionally, some minor grammatical errors, such as inconsistent verb tenses (eg, "were applied" vs. "apply"), should be corrected for consistency and readability.

General Comments

I have read the paper. Overall, the article is well structure and written. However, the topic is not novel several studies have been published on RGs. Furthermore, research gap needs to discuss in the introduction section more explicitly. How it advances our understanding of agriculture practices such as manure application and use of pesticides to influence AMR? In the introduction section, relevance of the research question to public health and food safety could be focused in detail. My further comments and suggestions are given below.

General comments;

Several studies used nano-particles (NPs) (<https://doi.org/10.1016/j.stress.2025.100917>; <https://doi.org/10.1016/j.plana.2024.100080>; <https://doi.org/10.3390/plants12132461>) for different bacterial pathogens control. Discuss this aspect in this article related to AMR.

In the introduction, clearly explain research gap, significance, relevance to human health, and significance of AMR study in tomato farming.

Recent studies needed to include particularly related to the combine effects of herbicides, pesticides, and manure on AMR in context of their impact on soil microbial communities in the Agri-ecosystems.

Microbial community shift, must be explain in context with AGRs in soil and plant leaves. In the results section p values in the figures and tables must be included consistently. Detail captions needed.

Discussion section needs deep literature in term of public health, food safety.

References style is inconsistent.

Specific comments

Abstract: Abstract need more streamline for quick reading, for example, summarize AGRs mention in this section without excessive detail.

M & M: Chemical dosages and application methods are not explained in sufficient detail. Provide concentration of chemical (glyphosate) and manure applied, compared to typical agricultural practices. In addition, why such concentration has been selected?

The study mentioned 64 experimental plots, but plots selection and randomization are not clear. How much replicates? How the treatments were assigned to each plot?

Why ANOVA and Tukey tests for statistical analysis were used while alternative tests are available. Microbial data processing and analysis need to explain.

Results: Explain microbial diversity indices in more detail treatment wise.

What are the biological mechanisms to shift the said microbial community?

Language and Grammar: The language is clear and technical. However, simplified sentences for better clarity. For instant, the sentence "Glyphosate treatment, alone or with manure, significantly increased total soil CARB by TP4" could be rephrased as "Glyphosate treatment, with or without manure, significantly increased CARB levels in the soil by TP4."

Additionally, some minor grammatical errors, such as inconsistent verb tenses (eg, "were applied" vs. "apply"), should be corrected for consistency and readability.

General Comments

I have read the paper. Overall, the article is well structure and written. However, the topic is not novel several studies have been published on RGs. Furthermore, research gap needs to discuss in the introduction section more explicitly. How it advances our understanding of agriculture practices such as manure application and use of pesticides to influence AMR? In the introduction section, relevance of the research question to public health and food safety could be focused in detail. My further comments and suggestions are given below.

Response: We thank for the reviewer for this comment. The manuscript was substantially revised based on the reviewer recommendations. Additional information was added in the introduction, materials and methods, and discussion sections to highlight the importance of this study in accordance with the current research gap, and the significance and relevance of AMR to human health and food safety. Details about the revisions made in the submitted manuscript are provide below.

General comments;

- **Several studies used nano particles (NPs) (<https://doi.org/10.1016/j.stress.2025.100917>; <https://doi.org/10.1016/j.plana.2024.100080>; <https://doi.org/10.3390/plants12132461>) for different bacterial pathogens control. Discuss this aspect in this article related to AMR.**

Response: We thank for the reviewer for this comment. The suggested studies were added in the discussion section to highlight to the antimicrobial potential of NP to mitigate pathogens, while minimizing the impact of agricultural practices on AMR burden (Line 568-585: “In addition to alternative manure management strategies, the use of emerging formulations and application strategies of antimicrobials to control plant diseases have shown promise as sustainable alternatives to conventional antimicrobials. For example, the use of silver, copper oxide, and zinc oxide nanoparticles have been reported to effectively suppress plant and soilborne bacterial pathogens, while reducing the risk of conventional antimicrobial overuse (1–6). Biological control agents (BCAs) are another alternative to conventional antimicrobials and in recent years new BCAs with improved efficacy as well as plant growth promotion are being explored (1) . Of particular interest is bacteriophage mediated biocontrol (1, 2). Bacteriophages are naturally occurring in the environment and have the generally recognized as safe (GRAS) status from the U.S. Food and Drug Administration (1). They have been successfully used in agriculture to manage destructive diseases such as bacterial spot and speck of tomato and pepper, citrus canker, and Pierce’s disease of grapevine. Lastly, the advancement of intelligent or smart spray technologies has dramatically improved the sustainability of antimicrobials through the reduction in chemical use and off-target run-off and drift, and slowing the development of AMR (1, 2). Incorporating these alternatives into integrated pest and resistance management frameworks can mitigate the spread of AMR and ultimately improving plant and public health. Of particular interest is bacteriophage mediated biocontrol (1, 2). Bacteriophages are naturally occurring in the environment and have the generally recognized as safe (GRAS) status from the

U.S. Food and Drug Administration (1). They have been successfully used in agriculture to manage destructive diseases such as bacterial spot and speck of tomato and pepper, citrus canker, and Pierce's disease of grapevine. Lastly, the advancement of intelligent or smart spray technologies has dramatically improved the sustainability of antimicrobials through the reduction in chemical use and off-target run-off and drift, and slowing the development of AMR (1, 2). Incorporating these alternatives into integrated pest and resistance management frameworks can mitigate the spread of AMR and ultimately improving plant and public health.”)

- **In the introduction, clearly explain research gap, significance, relevance to human health, and significance of AMR study in tomato farming.**

Response: We thank the reviewer for this comment. Additional information was added in the introduction section to highlight the research gap, significance, relevance to human health, and significance of AMR study in tomato farming (Line 110-123: “Together, these findings underscore the urgent need to disentangle how combined application of multiple agrochemical inputs act to reshape soil microbiomes and enhance the risk of AMR transmission from farm environments to human and plant ecosystems. Tomato farming provides a particularly significant model system for studying AMR dynamics. Tomatoes are one of the most widely consumed vegetables globally, often eaten raw, making them a direct exposure route for AMR bacteria, and resistance genes from soil, irrigation water, or manure to humans. As a high-value crop frequently treated with herbicides, fungicides, and sometimes antibiotics, tomato fields represent a hotspot for interactions between diverse agricultural inputs and microbial communities. The significance of studying AMR in tomato farming lies in its dual importance to safeguard food security and reduce risks to public health. Understanding how combined agricultural inputs shape soil and phyllosphere microbial communities, ARG, and the potential transfer of resistance from the environment to the food chain is critical for designing sustainable farming practices and mitigating human exposure.”)

- **Recent studies needed to include particularly related to the combined effects of herbicides, pesticides, and manure on AMR in context of their impact on soil microbial communities in the Agri ecosystems.**

Response: We thank for the reviewer for this comment. Additional information was added in the introduction section to highlight the combined effects of herbicides, pesticides, and manure on AMR in context of their impact on soil microbial communities in the Agri ecosystems (Line 104-109: “Despite growing evidence on the individual impacts of antibiotics, herbicides, and heavy metals on AMR, an important research gap remains. Most studies investigate single inputs, whereas real-world farming systems involve multiple inputs applied simultaneously. The interplay of these factors is increasingly recognized as a driver of AMR proliferation in agricultural ecosystems, particularly in intensively managed crop systems such as tomato cultivation (27–31).”).

- **Microbial community shift, must be explained in context with ARGs in soil and plant leaves.**

Response: We thank for the reviewer for this comment. Additional information about the interconnections between ARG and the agricultural ecosystem was added in the introduction section, Line 118-123: “The significance of studying AMR in tomato farming lies in its dual importance to safeguard food security and reduce risks to public health. Understanding how combined agricultural inputs shape soil and phyllosphere microbial communities, ARG, and the potential transfer of resistance from the environment to the food chain is critical for designing sustainable farming practices and mitigating human exposure.”

- **In the results section p values in the figures and tables must be included consistently. Detail captions needed.**

Response: We thank for the reviewer for this comment. The manuscript was revised to ensure p values were consistently mentioned in the text, figures and tables. Furthermore, caption have been revised to provide all the require information to understand the data presented in the designated figure and table.

- **Discussion section needs deep literature in term of public health, food safety.**

Response: We thank for the reviewer for this comment. Additional information was added in the discussion section to highlight the importance of our finding and their potential association with public health and food safety (Line 551-554: “Resistant bacteria and genes arising in crop production environments can be transferred through the food chain, irrigation water, or aerosols, ultimately threatening human health by limiting treatment options and increasing the risk of foodborne outbreaks linked to resistant pathogens (48)”); Line 599-602: “These findings are agronomically relevant, but also critical from a food safety perspective, as they may create favorable conditions for resistant bacteria to colonize edible plant surfaces, increasing the risk of human exposure through consumption of raw or minimally processed produce.”; Line 613-616: “ From a public health standpoint, understanding these interactions is essential because fungicide- and herbicide-driven resistance mechanisms may overlap with clinical resistance determinants, potentially narrowing therapeutic options against foodborne and opportunistic pathogens.”; Line 632-638: “This study aligns with the One Health framework, which emphasizes the interconnectedness of human, animal, and environmental health. Resistant bacteria and resistance genes originating in crop production systems do not remain confined to the agricultural environment but can disseminate through food products, water systems, and soil–human contact interfaces. This highlights the urgent need for integrated mitigation strategies that consider agricultural inputs, pathogen control innovations, and waste management practices in order to protect plant health, food safety and public health on a global scale.”)

- **References style is inconsistent.**

Response: We apologize for the oversight. The references were formatted using ASM Spectrum formatting style.

Specific comments

- **Abstract: Abstract need more streamline for quick reading, for example, summarize AGRs mention in this section without excessive detail.**

Response: We thank for the reviewer for this comment. The abstract was substantially revised as suggested. The abstract reads now as “Intensive pesticide use drives antimicrobial resistance (AMR) in agriculture, yet the effects of specific practices remain poorly understood. This study evaluated the impact of dairy manure and agrochemicals (glyphosate, copper, streptomycin, and propiconazole) on the composition of culturable AMR bacteria (CARB), AMR genes (ARGs; n=87) and microbiome in a processing tomato field (n=64 experimental plots). Glyphosate-treated plots harbored the lowest levels of CARB, but the highest prevalence of ARGs (especially tetA, tetB, OXA-50 and OXA-58) in the tomato leaves (P<0.05). Manure-treated plots had the highest level in CARB and ARGs in the soil and in tomato leaves (especially ACT-1, LAT, MIR, aadA1 and aphA6). The prevalence of multiple ARGs (IMP-12, ACT-1, DHA, MIR, MOX, OXA-58, OXA-60, ermB, oprj and oprm) was lower in streptomycin- or propiconazole- treated plots compared to non-treated plots. Shifts in the soil and leaf microbiome correlated with changes in ARG composition, particularly aminoglycoside-, fluoroquinolone-, and beta-lactamase-associated genes. These findings show that dairy manure, glyphosate, and propiconazole significantly alter the tomato field microbiome and ARG landscape, indicating that fungicide and herbicide applications may contribute to AMR development and dissemination similar to conventional antibacterial agents in agriculture ecosystems.” (Line 29-45).

- **M & M: Chemical dosages and application methods are not explained in sufficient detail. Provide concentration of chemical (glyphosate) and manure applied, compared to typical agricultural practices. In addition, why such concentration has been selected?**

Response: We thank for the reviewer for this comment. Additional information about the material and methods used in this study was revised. Line 149-154 : “Squeezed dairy manure was manually applied at a rate of 11.2 t/ha (standard application in Northeast Ohio) on the frozen soil of selected experimental plots at time point 2 (TP2; Table S1 and Figure S2). The manure was obtained from the CFAES-Wooster (previously Ohio Agricultural Research and Development Center) dairy farm (The Ohio State University, Wooster, OH). Neither the cows nor their associated manure received any antibiotic treatments.”; Line 155-168: “A post-emergent application of the herbicide glyphosate [Roundup Weathermax® Herbicide, Bayer Crop Science, St. Louis, MO; 48.8% potassium salt of glyphosate] was applied at a rate of 3.5 l/ha on the soil of selected experiment plots (one day after TP3) using a Farmall 140 field sprayer (International Harvester, Lisle, IL, U.S.; Table S1 and Figure S2). Antimicrobials (1.68 kg/ha of Kocide® 2000 DuPont Crop [Houston, TX; 53.8% copper hydroxide]; 200 µg/l of Harbour [ADAMA; Raleigh, NC; 22.4% streptomycin sulfate]; 0.35 l/ha of Propimax EC [Indianapolis, IN; active ingredient: propiconazole]) were applied weekly during the growing season directly to the seedling’s foliage of the selected experimental plots using a handheld 11.3 l pressurized backpack sprayer with an 80° flat fan nozzle (276 kPa), as recommended by the manufacturers (Table S1). Weeds were removed manually from the experimental plots using appropriate personal protective

equipment to avoid cross-contamination of the plots. Except for streptomycin, all agrochemicals were used according to the U.S. Environmental Protection Agency (EPA) registered labels. For streptomycin, off-label field applications were made for experimental purposes.”

- **The study mentioned 64 experimental plots, but plots selection and randomization are not clear. How much replicates? How the treatments were assigned to each plot?**

Response: We thank for the reviewer for this comment. Additional information about the material and method used in this study was revised. Line 138-147: “A total of 16 agricultural practice combinations were tested in this study. Each agricultural practices’ combination was randomized using a stratified strategy (four replicates per agricultural practice combination; total of 64 experimental plots; Figure S1). Each experimental plot was composed of three rows of 20 plants each. Before planting, manure, glyphosate, or manure plus glyphosate were applied once to experimental plots (Table S1 and Figure S2). After planting, experimental plots were treated on a weekly basis with streptomycin, copper, or propiconazole until harvest (Table S1 and Figure S2). Non-treated experimental plots were used as baseline to monitor the impact of the agricultural practices on the CARB, ARG and the microbiome.”

- **Why ANOVA and Tukey tests for statistical analysis were used while alternative tests are available. Microbial data processing and analysis need to explain.**

Response: We thank for the reviewer for this comment. The adequate statistical tests were used in this study to ensure an accurate detection of differences between the treatment groups or time points. ANOVA and Tukey tests were used to detect differences in bacterial load (CARB) between treatment groups during the study. Microbiome data were processed and analyzed using standard statistical packages (FASTQC, QIIME2, DADA2, SILVA database). Furthermore, PERMANOVA, Kruskal-Wallis’ test, Bootstrap Forest and ANCOM statistical methods were used for comparing groups and assessing significance of the microbiome data. Row-wise method and Pearson Product-Moment Correlation were used to identify correlations between continuous variables. Two-way clustering, principal component, and discriminant analyses were performed to estimate the variability observed between the agricultural practices’ combinations for a specific set of data. This information is now clarified in the method section (Line 300-331)

- **Results: Explain microbial diversity indices in more detail treatment wise.**

Response: We thank for the reviewer for this comment. Additional information about the microbial diversity trends observed between the treatments described in the manuscript (Line 488-497) were provided in greater details in the supplemental Table S4. Alpha diversity analysis based on Faith’s Phylogenetic Diversity (PD) and Shannon index revealed temporal and treatment-dependent differences in microbial community structure across soil and leaf samples. In soil samples, diversity fluctuated over time following the application of glyphosate and dairy manure (TP3–TP9). At TP4, both indices showed significant variation ($P < 0.01$), with the highest diversity observed in glyphosate-only plots (G) and the lowest in control plots (C). Similar patterns persisted at TP6 and TP7, where manure (M) and combined glyphosate-manure (GM)

treatments maintained higher microbial diversity compared to controls, whereas no significant differences were detected at TP8. By TP9, soil communities again showed distinct clustering, with decreasing diversity from G to M plots. In contrast, leaf-associated microbiomes exhibited generally lower diversity values, ranging from 3–31 for Faith's PD and 1.0–3.7 for Shannon index, but displayed clearer treatment effects. Leaf samples collected after glyphosate and manure applications (TP6–TP9) consistently showed higher diversity in manure-treated plots compared to controls. For antimicrobial-treated plots (copper, streptomycin, and propiconazole), soil alpha diversity decreased significantly at TP7 and TP8, with controls exhibiting greater richness than copper- and streptomycin-treated soils. By TP9, no significant differences were observed, suggesting a temporal recovery of the soil microbiome. Leaf microbiomes, however, remained more sensitive to treatments: at TP7, both Faith's PD and Shannon indexes were highest in control plants and lowest in streptomycin-treated leaves, with intermediate values in triazole- and copper-treated samples. Differences in leaf microbial diversity diminished over time, and by TP9, only minor distinctions persisted across treatments. Collectively, these results indicate that both herbicide-manure combinations and antimicrobial treatments temporarily altered microbial diversity, with stronger effects observed on phyllosphere communities than in soil environments.

- **What are the biological mechanisms to shift the said microbial community?**

Response: We thank for the reviewer for this comment. As mentioned in the revised manuscript, the shift in microbial community described in this study was multifactorial. Line 39-41: Shifts in the soil and leaf microbiome correlated with changes in ARG composition, particularly aminoglycoside-, fluoroquinolone-, and beta-lactamase-associated genes. Line 550-556, 561-568, 588-590, 595-599, and 603-609: Agricultural practices had a substantial impact on the microbial composition and associated ARG. We do not exclude the possibility that the physico-chemical properties of the soil contributed to these shifts over time.

- **Language and Grammar: The language is clear and technical. However, simplified sentences for better clarity. For instant, the sentence "Glyphosate treatment, alone or with manure, significantly increased total soil CARB by TP4" could be rephrased as "Glyphosate treatment, with or without manure, significantly increased CARB levels in the soil by TP4." Additionally, some minor grammatical errors, such as inconsistent verb tenses (eg, "were applied" vs. "apply"), should be corrected for consistency and readability.**

Response: We thank for the reviewer for this comment. The manuscript was revised to enhance the clarity of the information provided in this study.

Re: Spectrum02003-25R1 (**Dairy manure, glyphosate, and antimicrobials (copper, streptomycin and triazole) modulated the composition of antimicrobial resistance at the gene and microbial levels in a processing tomato field.**)

Dear Prof. Gireesh Rajashekara:

Your manuscript has been accepted, and I am forwarding it to the ASM production staff for publication. Your paper will first be checked to make sure all elements meet the technical requirements. ASM staff will contact you if anything needs to be revised before copyediting and production can begin. Otherwise, you will be notified when your proofs are ready to be viewed.

Sincerely,
Katharina Kujala
Editor
Microbiology Spectrum

Reviewer #2 (Comments for the Author):

No further comment